# Feature-Based Instance Neighbor Discovery: Advanced Stable Test-Time Adaptation in Dynamic World

**Qinting Jiang**
Shenzhen International Graduate School
Tsinghua University
jqt23@mails.tsinghua.edu.cn

**Chuyang Ye**
Courant Institute of Mathematical Sciences
New York University
chuyang.ye@nyu.edu

**Dongyan Wei**
CSDAI Department
Institut Polytechnique de Paris
dongyan.wei@ip-paris.fr

**Bingli Wang**
College of Information Engineering
Sichuan Agricultural University
wangbingli@stu.sicau.edu.cn

**Yuan Xue**
Shenzhen International Graduate School
Tsinghua University
yxue2021@gmail.com

**Jingyan Jiang**[*]
School of Artificial Intelligence
Shenzhen Technology University
jiangjingyan@sztu.edu.cn

**Zhi Wang**[*]
Shenzhen International Graduate School
Tsinghua University
wangzhi@sz.tsinghua.edu.cn

## Abstract

Despite progress, deep neural networks still suffer performance declines under distribution shifts between training and test domains, leading to a substantial decrease in Quality of Experience (QoE) for applications. Existing test-time adaptation (TTA) methods are challenged by dynamic, multiple test distributions within batches. We observe that feature distributions across different domains inherently cluster into distinct groups with varying means and variances. This divergence reveals a critical limitation of previous global normalization strategies in TTA, which inevitably distort the original data characteristics. Based on this insight, we propose **F**eature-based **I**nstance **N**eighbor **D**iscovery (FIND), which comprises three key components: Layer-Wise Feature Disentanglement (LFD), Feature-Aware Batch Normalization (FABN) and Selective FABN (S-FABN). LFD stably captures features with similar distributions at each layer by constructing graph structures; while FABN optimally combines source statistics with test-time distribution-specific statistics for robust feature representation. Finally, S-FABN determines which layers require feature partitioning and which can remain unified, thus enhancing the efficiency of inference. Extensive experiments demonstrate that FIND significantly outperforms existing methods, achieving up to approximately 30% accuracy improvement in dynamic scenarios while maintaining computational efficiency. The source code is available at `https://github.com/Peanut-255/FIND`.

---
[*]Corresponding author: wangzhi@sz.tsinghua.edu.cn, jiangjingyan@sztu.edu.cn

# 1   Introduction

The remarkable advancements in deep neural networks have not fully resolved the challenge of domain shift. When deployed in real-world scenarios, neural networks often encounter significant performance degradation as testing environments deviate from training conditions [1, 2, 3]. To mitigate these challenges, researchers have developed *test-time adaptation* (TTA) approaches, which enable models to dynamically adjust to new domains during inference without accessing the original training data or target domain labels [4].

Existing TTA approaches can be divided into two main categories: *test-time fine-tuning* [4, 3, 5, 6, 7, 8, 9, 10] and *test-time normalization* [11, 12, 13, 14]. The former methods adapt model parameters during inference [4, 3, 5, 15]. While effective, these approaches demand considerable computational resources. The latter methods, alternatively, focus on adapting batch normalization (BN) statistics to address distribution shifts. These approaches correct the statistics of source batch normalization (SBN) by leveraging either test-time batch normalization (TBN) or its variants during inference [3, 16].

The effectiveness of previous TTA methods primarily relies on the assumptions of ideal test conditions—test examples are homogeneous and originate from a single distribution over a period, which we refer to as the *static scenario* [4, 17, 3, 5]. However, **real-world data streams often exhibit more complex trends**: distribution shifts no longer gradually evolve over time; instead, multiple distinct distributions may emerge simultaneously, which we refer to as the *dynamic scenario*. For instance, due to variations in network conditions and device capabilities, clients may upload images of different compression rates. This results in the model receiving samples with varying levels of quality degradation that are drawn from multiple distinct distributions. Our investigation (Figure 2a) reveals that existing methods exhibit significant performance degradation in dynamic scenarios (an average accuracy drop of 15%). Surprisingly, the test-time fine-tuning approach not only fails to improve performance compared to test-time normalization but even decreases accuracy by approximately 0.4%. This finding highlights an essential revelation: in dynamic scenarios, mitigating performance degradation depends primarily on carefully adjusting normalization methods. We observe that feature distributions in dynamic scenarios naturally form distinct clusters with varying means and variances (Figure 3a). However, current TBN-based normalization approaches employ a global normalization strategy across all features, inducing interference between different distributions (Figure 1).

This brings us to a crucial insight: *in dynamic scenarios, effectively addressing performance degradation hinges on a "divide and conquer" normalization approach, which naturally raises two fundamental questions—how to divide and how to conquer?* To address these challenges, we start by performing preliminary experiments and motivation analyses to explore potential solutions (Section 3). We find that: 1) Adopting layer-wise feature partitioning instead of direct classification on raw input samples leads to superior efficacy. 2) Integrating generic knowledge from the source domain enhances normalization stability and compensates for missing generalizable patterns (e.g., class correlations) across partitioned groups.

Based on the above insights, we propose **FIND** (Feature-based Instance Neighbor Discovery), a fine-grained test-time normalization framework tailored for dynamic scenarios, comprising three key components: layer-wise feature disentanglement (LFD), feature-aware batch normalization (FABN) and selective FABN (S-FABN), as il-

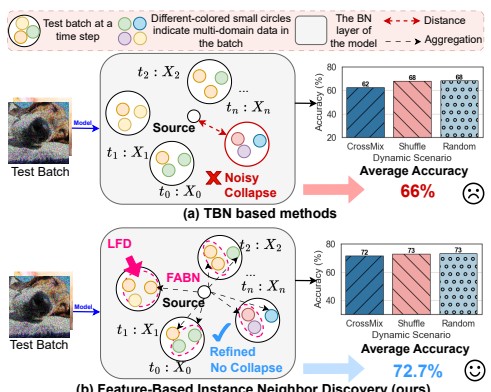

Figure 1: (a) In previous TBN-based methods, batch-level statistical values are computed holistically across all features, leading to erroneous normalization of cross-distribution features. (b) Our FIND framework introduces a layer-wise divide-and-conquer strategy for intra-batch feature processing, synergistically integrating source domain knowledge to achieve reliable normalization.

lustrated in Figure 1b. The core innovation of FIND lies in its layer-adaptive feature separation for multi-distribution samples and layer-specific distribution alignment, which precisely models the underlying statistical characteristics of each layer. For LFD, we establish graph-based feature modeling through the exploitation of inter-feature correlations, which enables systematic reorganization of

features across heterogeneous distributions. For FABN, we integrate the group-specific knowledge from each partitioned cluster with the generic knowledge derived from the statistics of the source domain to achieve robust normalization. Finally, we employ S-FABN to determine which layers require feature partitioning and which can remain unified, thereby enhancing inference efficiency. Our contributions can be summarized as follows:

- We introduce the pioneering test-time normalization framework specifically designed for realistic dynamic scenarios, addressing the limitations of current one-size-fits-all normalization approaches in practical applications.
- Our method employs instance-level statistics to identify and cluster features with similar distributions, achieving robust dynamic adaptation through the aggregation of group-specific knowledge and generic knowledge from the source domain.
- Our method demonstrates versatility across architectures equipped with batch normalization layers, including both transformer-based models (ViT) and classical convolutional networks (e.g., ResNet). Remarkably, we are the first BN-based method tested on ViT structures with outstanding performance in TTA.
- Comparative analysis on benchmark datasets validates our method's effectiveness, showing a 30% accuracy gain over current state-of-the-art solutions in dynamic scenarios.

## 2 Preliminary

Test-time adaptation addresses the challenge of model adaptation when deploying a pre-trained model in novel environments. Consider a deep neural network $f_\theta : \mathbf{x} \to y$ that has been trained on source domain data $\mathcal{D}_S$. During deployment, this model encounters a target domain containing $N$ unlabeled test samples, denoted as $\mathcal{D}_T$, $\{x_i\}_{i=1}^{N} \in \mathcal{D}_T$. While $\mathcal{D}_S$ and $\mathcal{D}_T$ may exhibit different distributions, they share the same output space for prediction tasks.

The fundamental objective of TTA is to enable real-time model adaptation using streaming test samples. This online adaptation process operates sequentially: at any time step $t$, the model processes a batch of instances $\mathbf{X}_t$, simultaneously performs adaptation and generates predictions $\hat{Y}_t$. A key characteristic of this setup is its streaming nature: when processing the subsequent batch $\mathbf{X}_{t+1}$ at step $t + 1$, the model must adapt and make predictions without retaining or accessing historical data $\mathbf{X}_{1 \to t}$. This constraint reflects real-world deployment scenarios where memory and computational resources are limited, and past data cannot be stored indefinitely.

In the TTA framework, target domain distributions evolve across time steps, with samples at each moment adhering to independent and identically distributed (i.i.d.) properties: $\mathbf{X}_t \in \mathcal{D}_t$ and $\mathcal{D}_t \neq \mathcal{D}_{t+1}$. However, following our earlier discussion, real-world applications commonly present situations where the inference process must handle mixed samples from varying distributions. To address this scenario, we model the target domain $\mathcal{D}_t$ at time step $t$ as a set comprising one or multiple distributions: $\mathcal{D}_t = \{\mathcal{D}_{t,1}, D_{t,2}, ..., D_{t,M}\}$, where $M \geq 1$.

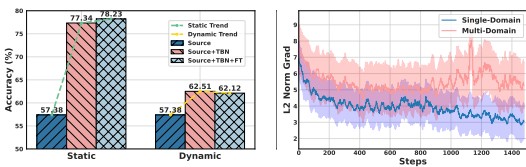

(a) TTN VS. FT     (b) Gradient of each step

Figure 2: (a) demonstrates the performance gains of test-time normalization (TTN) versus test-time fine-tuning (FT). (b) shows how gradient changes across fine-tuning steps under different scenarios.

This formulation, termed the *Dynamic* scenario, reflects that $\mathbf{X}_t$ may originate from either single or multiple distributions.

## 3 Observations of TTA in Dynamic Scenario

**Precise normalization is pivotal for effective TTA.** To disentangle the contributions of test-time normalization (TTN) and test-time fine-tuning, we conduct ablation studies by sequentially applying TBN and fine-tuning (SAR [7]) on the source model. As shown in Figure 2a, TBN demonstrates significantly greater contribution (5%-20% performance gains) compared to fine-tuning (-0.4%-1% gains by building upon TBN). Notably, fine-tuning introduces negative impacts in dynamic scenarios. Our gradient convergence analysis in Figure 2b reveals that in dynamic environments, conflicting

gradients from multi-distribution samples hinder convergence and degrade model performance, whereas static environments allow stable convergence beyond 1,000 adaptation steps. These findings establish the inefficacy of fine-tuning in dynamic scenarios, emphasizing the necessity of developing precise normalization strategies for robust performance enhancement.

**A divide-and-conquer strategy is essential in dynamic environments.** Figure 3a reveals intrinsic cluster patterns in BN layers when using domain as labels, where each cluster exhibits unique distribution characteristics (varying centers and variances). Conventional test-time normalization methods that compute global statistics inevitably distort feature distributions, as shown in Figure 3b and 3c: In static scenarios, different classes exhibit well-separated feature boundaries, whereas in dynamic scenarios, improper normalization induces feature coupling. This phenomenon motivates our core design principle: features from divergent distributions require separate normalization pipelines.

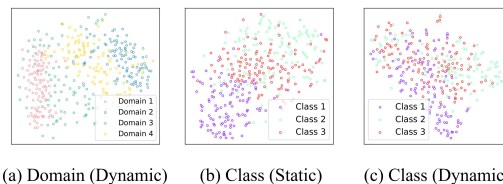

(a) Domain (Dynamic)  (b) Class (Static)  (c) Class (Dynamic)

Figure 3: Feature distributions across domains (a) and classes (b and c) under different scenarios. The domains are gaussian noise, zoom blur, snow, and pixelate from CIFAR100-C in (a). In (a) and (b), the feature distributions are clearly partitioned; in (c), they are entangled.

**Replace processing the input samples directly by layer-wise feature partitioning.** Direct partitioning input samples is infeasible due to missing domain labels and domain-class feature couplings. Inspired by Neyshabur et al. [18] and Lee et al. [8], who demonstrated that visual features inherently decompose into domain-relevant features (DRF, e.g., backgrounds) and class-relevant features (CRF, e.g., object structures), and that various network layers exhibit distinct preferences for these feature types, we propose layer-wise processing for BN layers. This strategy achieves: 1) reduced cross-feature interference during partitioning, and 2) fine-grained alignment with layer-specific normalization needs.

**Enhance normalization stability via generic knowledge from training data.** When the "divide-and-conquer" strategy is implemented, the features of the BN layer will be divided into multiple clusters. The sizes of these clusters are much smaller than the original batch sizes, resulting in biases in the distribution of class-relevant features. As shown in Figures 4a and 4b, while TBN exhibits sensitivity to batch size and domain quantity, SBN maintains stable performance across these variables. This robustness stems from SBN's integration of generic knowledge (e.g., class-relevant knowl-

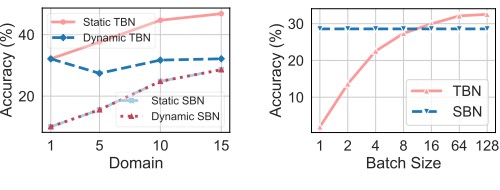

(a) DN vs. accuracy  (b) BS vs. accuracy

Figure 4: (a) and (b) display the inference accuracy with different domain numbers (DN) and batch sizes (BS). Static and Dynamic represent different scenarios.

edge)—comprehensive statistics derived from large-scale training data. These domain-shareable characteristics facilitate stable normalization by preventing the loss of generic features in clusters. A more specific analysis on the introduction of SBN in TBN is in Appendix M.

## 4 Method: FIND

As shown in Figure 5, the proposed FIND involves three novel designs, including layer-wise feature disentanglement (LFD), feature-aware batch normalization (FABN) and selective FABN (S-FABN). Here we introduce them in the following parts.

### 4.1 Layer-Wise Feature Disentanglement

In previous methods, the monolithic statistical computation—applied uniformly across all batch samples—inevitably skews feature representations toward erroneous distributions, particularly under dynamic multi-domain shifts, thereby compromising the overall reliability and accuracy of model inference during test-time adaptation. Our analysis reveals that a divide-and-conquer strategy is essential for normalizing multi-domain samples. However, directly partitioning input samples faces

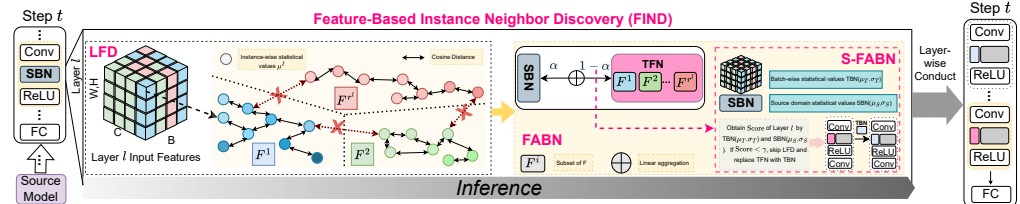

Figure 5: Overview of the proposed FIND (FIND*). We design layer-wise feature disentanglement (LFD) to divide multi-distribution feature maps within each layer into different subsets. Features in each subset have similar cosine distances, ensuring distribution consistency. Next, we combine the test-feature-specific normalization (TFN) statistics calculated for each subset with SBN statistics to improve normalization stability. Finally, layers requiring LFD partitioning can be selected through sensitivity score to enhance efficiency.

practical limitations: For supervised clustering algorithms, the lack of relevant domain labels and domain information poses challenges; for some unsupervised clustering algorithms, it is difficult to determine the number of clusters, while category features, domain features, and other characteristics in the samples are inherently entangled. Disentangling these features may require additional training steps. Inspired by [18] and our observations that BN layers exhibit layer-specific feature preferences (e.g., shallow layers focus on edges/textures while deeper layers prioritize semantic structures), we propose layer-wise feature disentanglement (LFD). Unlike sample-level segregation, this neighbor-search-based approach achieves fine-grained feature separation tailored to each layer's distributional characteristics.

For a certain BN layer in the network, we analyze the feature map $F \in \mathbb{R}^{B \times C \times H \times W}$ of all samples at this layer, where $B$ represents the batch dimension, $C$ denotes the number of channels, and spatial dimensions are spatial dimensions are represented by $L$ ($L = HW$). Central to our approach is the identification and re-grouping of feature maps that exhibit similar feature characteristics. This grouping process is performed based on instance-level statistical representations $\mu^I \in \mathbb{R}^C$, which capture the essential features of each sample in the batch. The computation of these instance-level means $\mu^I$ and the methodology for assessing feature similarity between samples can be formulated as follows:

$$\mu^I_{i,c} = \frac{1}{L} \sum_L F_{i;c;L}, \tag{1}$$

$$\mathrm{Sim}(i,j) = \frac{\mu^I_{i,c} \cdot \mu^I_{j,c}}{\left\| \mu^I_{i,c} \right\| \left\| \mu^I_{j,c} \right\|}, \tag{2}$$

where $F_{i;c;L}$ and $F_{j;c;L}$ represent feature maps of sample $i$ and sample $j$. Utilizing the $\mathrm{Sim}$ metric to measure the similarities between instance-level statistical representations across samples, LFD partitions the feature maps $F$ of all samples into distinct subsets at this BN layer.

Drawing inspiration from FINCH [19], our LFD approach partitions $F$ based on first-neighbor relationships of instance-level statistics. For each BN layer, we compute similarity metric $\mathrm{Sim}$ between different $\mu^I$ pairs, where $\mathrm{Sim}$ quantifies the feature affinity between instances at the current layer. Each $\mu^I$ identifies its first neighbor by selecting the instance with the highest similarity score. Following this first-neighbor initialization, LFD constructs an adjacency matrix as follows:

$$\mathrm{First}(i,j) = \begin{cases} 1, & \text{if } a^1_i = j \text{ or } a^1_j = i \text{ or } a^1_i = a^1_j \\ 0, & \text{otherwise} \end{cases}, \tag{3}$$

where each sample's feature map at the layer constitutes the node of the adjacency matrix. $a^1_i$ represents the first neighbor of the $i - th$ node. After obtaining the adjacency matrix $\mathrm{First}$, LFD proceeds with feature map $F$ partitioning. By traversing the adjacency matrix $\mathrm{First}$, nodes $i$ and $j$ are assigned to the same connected component when $\mathrm{First}(i,j) = 1$. This operation reconstructs $F$ into a graph structure with multiple connected components, thereby grouping features exhibiting high similarity measures. The detailed algorithm is provided in Appendix A. The analysis of clusters of each layer by LFD is provided in Appendix H.

## 4.2 Feature-Aware Batch Normalization

After partitioning, the original feature map $F$ in the BN layer is divided into $r$ subsets, i.e., $F = \left\{F^1, F^2, F^3, \ldots, F^r\right\}$, where $F^i \in \mathbb{R}^{b \times C \times L}$ and $b$ represents the number of instance-wise feature maps included in that set. Similar to computing batch-wise statistical values, we calculate the test-feature-specific-normalization (TFN) statistical values $\mu_F^i$ and $\sigma_F^i$ for each subset:

$$\mu_F^i = \frac{1}{bL} \sum_{b,L} F_{b;c;L}^i, \ \sigma_F^i = \sqrt{\frac{1}{bL} \sum_{b,L} \left(F_{b;c;L}^i - \mu_F^i\right)^2}. \tag{4}$$

Based on the preceding analysis, TFN computations enable the capture of domain-specific feature distributions within each BN layer. However, since TFN operates on smaller subsets compared to the original batch size, the missing or incomplete representation of generic features (e.g., insufficient class diversity in subsets) in these subsets may introduce statistical bias to TFN estimations. Therefore, we introduce SBN (generic knowledge) to supplement TFN. We thus propose Feature-Aware Batch Normalization (FABN), which utilizes SBN to correct the TFN statistics:

$$\mu_{\text{FABN}}^i = \alpha\mu_s + (1 - \alpha)\mu_F^i, \tag{5}$$

$$\sigma_{\text{FABN}}^i{}^2 = \alpha\sigma_s^2 + (1 - \alpha)\sigma_F^i{}^2, \tag{6}$$

$\alpha$ determines the relative contributions of each component. This hybrid approach leverages pre-trained model knowledge through SBN while utilizing TFN to mitigate domain-specific feature perturbations. The final FABN output can be expressed as:

$$\text{FABN}^i\left(\mu_{\text{FABN}}^i, \sigma_{\text{FABN}}^i\right) = \varphi \cdot \frac{\left(F_{;c;}^i - \mu_{\text{FABN}}^i\right)}{\sqrt{\sigma_{\text{FABN}}^i{}^2 + \varepsilon}} + \beta, \quad 1 \leq i \leq r, \tag{7}$$

where $\varphi$ and $\beta$ represent the affine parameters of the BN layer, and $\varepsilon$ is a small bias to prevent division by zero.

## 4.3 Selective Feature-Aware Batch Normalization (S-FABN)

As noted earlier, different layers exhibit varying preferences for features, meaning layers with minimal focus on domain characteristics remain insensitive to domain shifts. Consequently, feature map partitioning becomes unnecessary for these layers since their feature distributions across domains show minimal divergence and align closely with the source domain. To quantify the distribution gap, we calculate the KL divergence between source domain $\mathcal{D}_S$ and current target domain $\mathcal{D}_T$ for each BN layer:

$$\mathcal{KL}(\mathcal{D}_T \parallel \mathcal{D}_S) = \frac{\sigma_T^2 + (\mu_T - \mu_S)^2}{2\sigma_S^2} + \ln\left(\frac{\sigma_S}{\sigma_T}\right) - \frac{1}{2}, \tag{8}$$

where $\mathcal{KL} \in \mathbb{R}^C$, $\mu_T = \frac{1}{BL} \sum_{B,L} F_{B;c;L}$ and $\sigma_T = \sqrt{\frac{1}{BL} \sum_{B,L} \left(F_{B;c;L} - \mu_T\right)^2}$. We calculate the mean KL divergence $\mathcal{KL}_{mean}$ across all channels in the current BN layer based on $\mathcal{KL}$. However, averaging across all channels risks diluting critical signals from highly sensitive ones. We address this by:

$$\text{Score} = (1 + \frac{1}{1 + e^{-\mathcal{KL}_{st}}})\mathcal{KL}_{mean}, \tag{9}$$

where $\mathcal{KL}_{st}$ is the standard deviation of KL divergence in the current BN layer. The introduction of $\mathcal{KL}_{st}$ as a weighting factor mitigates potential oversight of highly sensitive channels.

In this process, we introduce no additional data but conduct a cold-start phase. Specifically, we leverage the initial batches fed into the model to compute the average Score for each layer, followed by Score normalization. Throughout this phase, the divide-and-conquer strategy remains applied to all layers. Upon obtaining the scores, partitioning is deactivated for layers exhibiting lower sensitivity. We set a threshold $\gamma$. If Score $\geq \gamma$, we retain the FABN. Otherwise, we deactivate the partitioning and replace the TFN component in FABN with $(\mu_T, \sigma_T)$.

We denote the selectively partitioned FIND as FIND*. We conduct sensitivity analysis on $\gamma$ in the experiments. The details of the cold-start process and the sensitivity analysis of $\gamma$ are given in Appendix G and Appendix I.

# 5 Experiments

## 5.1 Experimental Setup

We evaluate our proposed FIND using the Test-Time Adaptation Benchmark (TTAB) [20]. All reported results represent averages across three independent runs with different random seeds. Full implementation details and experimental configurations are provided in the Appendix.

**Environment and Hyperparameter Configuration.** Our experiments were conducted using an NVIDIA RTX 4090 GPU and a V100 GPU with PyTorch 1.10.1 and Python 3.9.7. For our FABN module in FIND, we set the aggregation parameter $\alpha$ to 0.8. We set the threshold $\gamma$ to 0.1 for FIND*. A detailed analysis of $\alpha$, $\gamma$ and the cold-start phase of FIND* is available in the Appendix. The details of hyperparameter settings are provided in Appendix F.

**Baselines.** We consider the following baselines: (1) **Test-time fine-tune.** SAR [7], EATA [21], DeYO [8], TENT [4], NOTE [3], RoTTA [5] and ViDA [6]. (2) **Test-time normalization.** TBN [22], $\alpha$-BN [23], and IABN [3]. We evaluate all methods under the online test-time adaptation (TTA) protocol, where source training data is inaccessible. Detailed experimental configurations are provided in Appendix C.

Following standard protocols [20, 3], we conduct experiments with a test batch size of 64 and one adaptation epoch. Method-specific hyperparameters are adopted from their respective published configurations [20].

**Datasets.** We evaluate on three corrupted datasets from the TTA benchmark [20]: CIFAR10-C (10-C), CIFAR100-C (100-C), and ImageNet-C (IN-C) [24]. Similar to previous studies [20], all experiments use severity level 5 corruptions and ResNet-50 [25] models pre-trained on their respective clean datasets. **To demonstrate the transferability of our approach, we extend our evaluation to transformer-based architectures, specifically using EfficientViT [26] as an additional backbone. We are the first BN-based method tested on ViT.** Details of datasets are provided in Appendix B.

**Scenarios.** In contrast to existing approaches that assume static data patterns, we shift our attention to TTA in dynamic data patterns. In our experiments, we employed three scenarios based on dynamic data patterns: **CrossMix.** Each batch contains samples from multiple domains (15 domains). **Shuffle.** Batches alternate between containing samples from a single domain and multiple domains. **Random.** The number of domains represented in each batch is entirely stochastic.

All three scenarios involve abrupt distribution shifts between consecutive time steps. Detailed scenario specifications are provided in Appendix E.

## 5.2 Performance Comparison under Dynamic Scenarios

Table 1 presents the comparative results of all test-time adaptation methods under dynamic scenarios. Our method consistently outperforms existing approaches in all scenarios. Most importantly, in the CrossMix scenario, we achieve accuracy gains of 17% and 7% over RoTTA (worst baseline) and DeYO (best baseline), respectively. This superior performance demonstrates our method's effectiveness in handling multi-distribution batch data, attributed to FABN's ability to effectively partition features into distribution-specific subsets while maintaining their statistical integrity. The performance advantages persist in both Random and Shuffle scenarios, where we surpass the strongest baseline ($\alpha$-BN) by 3% in three datasets. These results validate our method's robustness across various distribution shift patterns, regardless of whether the test batches contain single, multiple, or transitioning distributions. Notably, there is almost no performance gap between FIND* and FIND, which better reflects each layer's individual sensitivity to distribution shifts. For insensitive layers, avoiding feature partitioning can reduce additional time consumption. The results under different domain scales of a batch are provided in Appendix K. The results in other scenarios are provided in Appendix N, P and Q respectively.

## 5.3 Performance Comparison under Transformer Backbone

We use EfficientViT [26] as an additional backbone. EfficientViT is an architecturally optimized vision transformer that replaces the conventional layer normalization with batch normalization layers to achieve computational efficiency. We are the first BN-based method tested on ViT. Table 2

presents the comparative results. Our method consistently outperforms existing approaches across all scenarios. Most notably, in the CrossMix scenario, we achieve accuracy gains of 10% and 5% over NOTE (worst baseline) and Source (best baseline), respectively. The performance advantages persist in both Random and Shuffle scenarios, where we surpass all the baselines. It demonstrates our method has versatility across architectures equipped with batch normalization layers, including both transformer-based models (ViT) and classical convolutional networks (e.g., ResNet).

Table 1: Comparison with state-of-the-art methods on CIFAR10-C, CIFAR100-C, and ImageNet-C datasets (**corruption severity 5, batch size 64**). Results show **accuracy (%)** across CrossMix, Random, and Shuffle scenarios using **ResNet-50**. Best and second-best results are shown in bold and underlined, respectively.

| Method | Venue | CrossMix | | | | Random | | | | Shuffle | | | | |
| | | 10-C | 100-C | IN-C | Avg. | 10-C | 100-C | IN-C | Avg. | 10-C | 100-C | IN-C | Avg. | Avg-All |
|---|---|---|---|---|---|---|---|---|---|---|---|---|---|---|
| Source | CVPR16 | 57.39 | 28.59 | 25.64 | 37.21 | 57.38 | 28.58 | 25.93 | 37.30 | 57.38 | 28.58 | 25.80 | 37.25 | 37.25 |
| *TEST-TIME FINE-TUNE* | | | | | | | | | | | | | | |
| TENT | CVPR21 | 62.77 | 31.57 | 18.45 | 37.60 | 71.50 | 40.96 | 23.15 | 45.20 | 71.56 | 40.73 | 23.78 | 45.36 | 42.72 |
| EATA | ICML22 | 61.97 | 32.74 | 19.68 | 38.13 | 68.25 | 42.15 | 24.26 | 44.89 | 67.83 | 41.37 | 24.50 | 44.57 | 42.53 |
| NOTE | NIPS22 | 63.03 | 32.96 | 17.44 | 37.81 | 62.81 | 33.19 | 21.64 | 39.21 | 65.34 | 35.12 | 22.98 | 41.15 | 39.39 |
| SAR | ICLR23 | 61.70 | 31.45 | 18.65 | 37.27 | 71.04 | 40.92 | 23.62 | 45.19 | 71.49 | 39.58 | 23.50 | 44.86 | 42.44 |
| RoTTA | CVPR23 | 43.70 | 24.05 | 21.85 | 29.87 | 48.68 | 23.80 | 20.39 | 30.96 | 54.79 | 29.29 | 22.71 | 35.60 | 32.14 |
| ViDA | ICLR24 | 61.97 | 32.14 | 18.52 | 37.54 | 67.96 | 39.48 | 23.33 | 43.59 | 67.96 | 39.48 | 23.06 | 43.50 | 41.54 |
| DeYO | ICLR24 | 68.85 | 30.43 | 19.13 | 39.47 | **75.63** | 36.67 | 24.32 | 45.54 | **75.62** | 35.45 | 25.21 | 45.43 | 43.48 |
| *TEST-TIME NORMALIZATION* | | | | | | | | | | | | | | |
| TBN | ICML20 | 61.96 | 32.12 | 18.72 | 37.60 | 67.75 | 39.16 | 22.99 | 43.30 | 67.63 | 39.22 | 23.35 | 43.40 | 41.43 |
| $\alpha$-BN | arXiv20 | 62.41 | 33.22 | 21.92 | 39.18 | 69.88 | 41.59 | 27.57 | 46.35 | 69.87 | 41.60 | 27.78 | 46.42 | 43.98 |
| IABN | NIPS22 | 62.63 | 24.54 | 9.73 | 32.30 | 64.59 | 26.40 | 10.75 | 33.91 | 64.59 | 26.40 | 10.79 | 33.93 | 33.38 |
| FIND | Proposed | **71.54±0.2** | 39.75±0.2 | 29.21±0.0 | 46.83 | 73.09±0.2 | 42.56±0.2 | 30.12±0.2 | 48.59 | 72.74±0.1 | 42.87±0.3 | 30.00±0.2 | 48.54 | 47.87 |
| FIND* | Proposed | 70.75±0.0 | **40.48±0.1** | **30.33±0.1** | **47.19** | 73.68±0.0 | **43.05±0.0** | **30.62±0.0** | **49.12** | 73.60±0.0 | **43.88±0.2** | **30.24±0.4** | **49.23** | **48.50** |

Table 2: Comparison with state-of-the-art methods on CIFAR10-C, CIFAR100-C, and ImageNet-C datasets (**corruption severity 5, batch size 64**). Results show **accuracy (%)** across CrossMix, Random, and Shuffle scenarios using **EfficientViT-M5**. Best and second-best results are shown in bold and underlined, respectively.

| Method | Venue | CrossMix | | | | Random | | | | Shuffle | | | | |
| | | 10-C | 100-C | IN-C | Avg. | 10-C | 100-C | IN-C | Avg. | 10-C | 100-C | IN-C | Avg. | Avg-All |
|---|---|---|---|---|---|---|---|---|---|---|---|---|---|---|
| Source | CVPR16 | 74.57 | 42.87 | 26.05 | 47.83 | 74.62 | 43.14 | 27.39 | 48.38 | 74.62 | 43.06 | 27.36 | 48.35 | 48.19 |
| *TEST-TIME FINE-TUNE* | | | | | | | | | | | | | | |
| TENT | CVPR21 | 74.22 | 42.54 | 20.37 | 45.71 | 77.98 | 44.70 | 23.58 | 48.75 | 77.97 | 44.86 | 23.67 | 48.83 | 47.76 |
| EATA | ICML22 | 74.50 | 41.79 | 20.75 | 45.68 | 77.93 | 45.28 | 24.57 | 49.26 | 77.88 | 45.47 | 24.11 | 48.83 | 47.92 |
| NOTE | NIPS22 | 68.45 | 36.42 | 20.10 | 41.66 | 67.05 | 35.19 | 18.95 | 40.40 | 65.56 | 34.24 | 18.50 | 39.43 | 40.50 |
| SAR | ICLR23 | 75.17 | 41.39 | 20.18 | 45.58 | 78.29 | 44.50 | 23.62 | 48.80 | 77.80 | 45.76 | 23.43 | 49.00 | 47.79 |
| RoTTA | CVPR23 | 75.84 | 43.35 | 20.10 | 46.43 | 74.52 | 42.58 | 20.99 | 46.03 | 76.07 | 44.09 | 23.63 | 47.93 | 46.79 |
| ViDA | ICLR24 | 75.05 | 41.37 | 12.45 | 42.96 | 78.13 | 44.73 | 14.30 | 45.72 | 78.20 | 44.67 | 15.51 | 46.13 | 44.94 |
| DeYO | ICLR24 | 75.10 | 42.50 | 20.35 | 45.98 | 78.01 | 44.54 | 24.45 | 49.00 | 78.35 | 44.65 | 20.83 | 47.94 | 47.64 |
| *TEST-TIME NORMALIZATION* | | | | | | | | | | | | | | |
| TBN | ICML20 | 74.94 | 41.26 | 20.06 | 45.42 | 77.73 | 43.84 | 23.10 | 48.22 | 77.65 | 44.75 | 22.87 | 48.42 | 47.35 |
| $\alpha$-BN | arXiv20 | 75.50 | 43.90 | 22.97 | 47.46 | 78.47 | 46.95 | 28.44 | 51.29 | 78.94 | 46.92 | 28.55 | 51.47 | 50.07 |
| IABN | NIPS22 | 68.47 | 37.63 | 20.21 | 42.10 | 67.08 | 34.66 | 19.20 | 40.31 | 66.68 | 35.04 | 18.52 | 40.08 | 40.83 |
| FIND | Proposed | 78.60±0.2 | 48.69±0.2 | 29.23±0.0 | 52.17 | 79.54±0.1 | 48.75±0.2 | 30.94±0.0 | 53.08 | 79.58±0.1 | 48.47±0.1 | 30.95±0.2 | 53.00 | 52.75 |
| FIND* | Proposed | **78.72±0.2** | **49.30±0.4** | **30.05±0.1** | **52.69** | **79.61±0.1** | **49.10±0.3** | **31.17±0.2** | **53.29** | **79.60±0.2** | **48.89±0.2** | **31.24±0.3** | **53.24** | **53.07** |

## 5.4 Efficiency Analysis

To evaluate the efficiency of different methods, we compare their

Table 3: Comparison of memory (GB) and latency (s) of different methods.

| Overhead | TBN | $\alpha$-BN | TENT | IABN | DeYO | RoTTA | SAR | EATA | ViDA | NOTE | FIND | FIND* |
|---|---|---|---|---|---|---|---|---|---|---|---|---|
| Latency (s) | 0.07 | 0.07 | 0.17 | 0.22 | 0.23 | 0.60 | 0.30 | 0.17 | 5.62 | 1.84 | **0.21** | **0.15** |
| Memory (GB) | 0.86 | 0.86 | 5.59 | 1.73 | 6.20 | 12.72 | 5.59 | 5.64 | 7.02 | 10.94 | **1.44** | **1.21** |

memory usage and latency on ImageNet-C using a V100 GPU and ResNet-50. Table 3 shows that our proposed methods, **FIND** and **FIND\***, achieve significantly lower memory usage and latency compared to most of the baselines.

As shown in Table 3, our method demonstrates strong computational efficiency, ranking top-3 in inference speed (30× faster than ViDA) and memory usage (12× more efficient than RoTTA), confirming its practical deployability.

## 5.5 Sensitivity analysis of $\gamma$

We compress the score of each layer in FIND* to the range $[0, 1]$ through normalization, with score proximity to 0 indicating lower distribution shift sensitivity of the layer. As shown in Figure 6, when $\gamma$ varies within 0-0.1, accuracy fluctuation remains below 1% while inference efficiency improves by 17%-32%, demonstrating robust model performance. When $\gamma$ exceeds 0.1, accuracy drops by 2%-6% with unstable model behavior. Therefore, setting $\gamma$ between 0-0.1 achieves inference acceleration while preserving model performance.

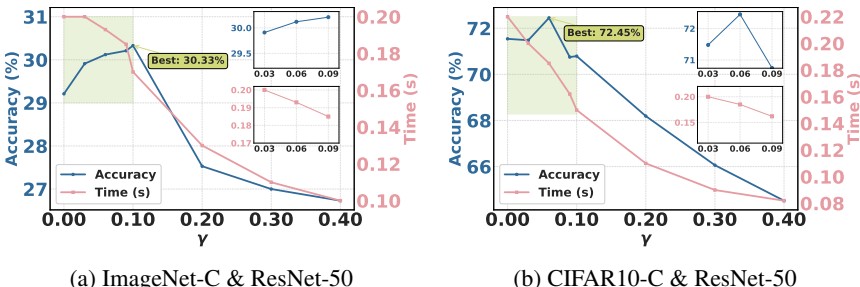

(a) ImageNet-C & ResNet-50  (b) CIFAR10-C & ResNet-50

Figure 6: **(a)** and **(b)** demonstrate the sensitivity of model inference performance (Accuracy (%)) to $\gamma$ variations, along with the associated changes in inference efficiency (Time (s)).

## 5.6 Performance Comparison Between FIND (Ours) and Other Clustering Algorithms

Table 4 displays the performance of LFD in our method and other clustering algorithms on ImageNet-C. **Requires no predefined cluster count:** HDBSCAN [27], DB-SCAN [28], RccCluster [29], and DSets-DBSCAN [30] are density-based algorithms, while Agglomerative [31] and Birch [32] are hierarchy-based. **Requires predefined cluster:** K-means [33]. Our method demonstrates superior performance and computational efficiency compared to alternative clustering approaches. This demonstrates that LFD can effectively aggregate distribution-relevant features with high similarity, thereby providing a cleaner representation. In conclusion, FIND achieves fully auto-

Table 4: Comparison of substituting our LFD in FIND with other clustering methods. The model is ResNet-50.

| Method | Acc (%) | Time (s) | Mem (GB) |
|---|---|---|---|
| HDBSCAN | 24.66 | 0.50 | 1.64 |
| Agglomerative | 25.77 | 0.33 | 1.56 |
| Birch | 26.09 | 2.12 | 2.06 |
| RccCluster | 25.30 | 9.51 | 1.66 |
| DSets-DBSCAN | 25.13 | 2.05 | 1.62 |
| DBSCAN | 25.71 | 1.34 | 2.01 |
| K-means | 26.71 | 9.74 | 15.91 |
| FIND (Ours) | **29.21** | **0.21** | **1.44** |

matic perception and dynamic grouping. Other clustering algorithms depend on hyperparameters or can only partition the feature space globally, lacking the capability for fine-grained perception in complex scenarios.

## 5.7 Performance under Different Batch Size

Figure 7 demonstrates our method's stability across varying batch sizes in the CrossMix scenario. While baseline methods exhibit significant performance degradation with smaller batches, particularly on CIFAR100-C where accuracy stabilizes only beyond batch size 64, our approach maintains consistent performance regardless of batch size. This batch-size invariance highlights our method's robustness.

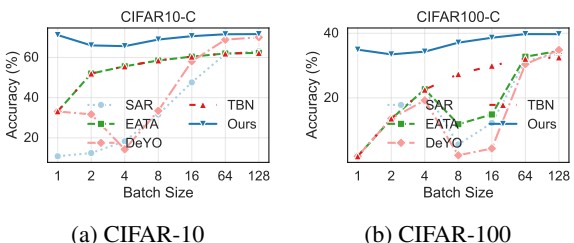

(a) CIFAR-10  (b) CIFAR-100

Figure 7: Batch size sensitivity analysis. Our approach demonstrates batch-size invariant performance.

## 5.8 Ablation Study on $\alpha$

We conducted two sets of ablation studies on $\alpha$ with $\gamma = 0.1$ in FABN, separately for clustered and non-clustered layers (Table 6). We observe that in both cases, performance is better when

Table 6: Ablation study of $\alpha$ for clustered layers (CL) and non-clustered layers (NCL) in the CrossMix scenario, where $\alpha$ for one layer type is varied while $\alpha$ for the other is fixed at 0.8. The dataset is CIFAR10-C (**severity level 5, batch size 64**) and the model is ResNet-50.

| | $\alpha$ (CL) | | | | | | $\alpha$ (NCL) | | | | | |
| | **0** | **0.2** | **0.4** | **0.6** | **0.8** | **1** | **0** | **0.2** | **0.4** | **0.6** | **0.8** | **1** |
|---|---|---|---|---|---|---|---|---|---|---|---|---|
| Acc (%) | 57.48 | 62.98 | 66.81 | 69.49 | 70.75 | 60.39 | 67.20 | 67.47 | 68.29 | 69.36 | 70.75 | 70.80 |

the SBN ratio is higher (>0.5). Clustered layers are sensitive to the SBN ratio (13% change in accuracy), while non-clustered layers are not (3% change in accuracy). The ablation study on $\alpha$ provides strong evidence supporting several conclusions in our paper: 1. The generic knowledge (SBN) dominates during inference. 2. The sensitivity of $\alpha$ is much lower in layers that do not require clustering compared to those that do, because TFN only provides domain-specific distributions rather than the full generic knowledge distribution (as discussed in Section 3, Observation 4).

### 5.9 Ablation Study on FIND

We conducted the ablation study on the components of FIND, which consists of three key modules: LFD, FABN, and S-FABN. LFD performs hierarchical feature partitioning to derive new normalization statistics, referred to as TFN. FABN complements TFN by incorporating SBN to supply the missing generic features. Table 5 reveals that neither SBN nor TFN alone achieves optimal results. SBN lacks relevant distribution information from the target domain, while TFN's small-batch

Table 5: Ablation study on corrupted datasets using ResNet-50 (**severity level 5, batch size 64**) in the CrossMix scenario.

| Method | 10-C | 100-C | IN-C | Avg. |
|---|---|---|---|---|
| SBN | 57.39 | 28.59 | 25.64 | 37.21 |
| TFN | 45.78 | 18.90 | 10.65 | 25.11 |
| SBN+TFN (FIND) | **71.54** | 39.75 | 29.21 | 46.83 |
| FIND* | 70.75 | **40.48** | **30.33** | **47.19** |

computation limits its ability to capture generic feature distributions. Our proposed FIND leverages the complementary strengths of both approaches, thereby enhancing overall performance.

## 6 Related Work

**Online TTA and Continual TTA.** Test-time adaptation (TTA), a form of unsupervised domain adaptation, enables model adaptation without source data. Initial work by Schneider et al.[34] demonstrated the effectiveness of updating batch normalization statistics during inference. Subsequent approaches incorporated backpropagation-based optimization, with Wang et al.[4] focusing on batch normalization parameters through entropy minimization, while Zhang et al. [35] extended this to full model optimization with test-time augmentation. Continual TTA addresses sequential domain shifts in real-world deployments. While TENT [4] can handle continuous adaptation, it risks error accumulation. COTTA [17] specifically targets this challenge through weighted averaging and restoration mechanisms. EATA [21] further enhances adaptation stability by incorporating entropy-based sample filtering and Fisher regularization.

**TTA in Dynamic Wild World.** Real-world test data often exhibits complex characteristics, including data drift and mixed or imbalanced distributions. SAR [7] examines model degradation under such wild data, NOTE [3] tackles non-i.i.d. distributions through IABN and balanced sampling, and DeYo [8] leverages feature decoupling for enhanced category learning.

However, TTA in dynamic environments remains largely unexplored. We present the first dedicated analysis of distribution shifts through BN statistics correction, introducing a backward-free adaptation framework. More details can be found in Appendix D.

## 7 Conclusion

In this paper, we propose FIND, a divide-and-conquer strategy to enhance TTA performance in dynamic scenarios. FIND combines LFD for feature map partitioning via instance normalization statistics, and FABN for consistent representation through SBN aggregation. Our approach demonstrates robust performance across diverse distribution shifts in dynamic data streams compared with existing methods. More discussions are in Appendix U.

# 8 Acknowledgements

This work is supported in part by the National Key Research and Development Project of China (Grant No. 2023YFF0905502), National Natural Science Foundation of China(Grant No. 92467204 and 62472249), Shenzhen Science and Technology Program (Grant No. JCYJ20220818101014030 and KJZD20240903102300001) and Natural Science Foundation of Top Talent of SZTU (Grant No. GDRC202413).

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

# Contents of Appendix

# A    The Layer-Wise Feature Disentanglement Algorithm

Algorithm 1 is the proposed layer-wise feature disentanglement (LFD) Algorithm in this paper.

The identification of semantically similar samples is achieved through cosine similarity computations between feature map representations at specific batch normalization layers. This similarity-based approach enables precise detection of nearest neighbors that share comparable distributional characteristics. By constructing an adjacency matrix (detailed in Equation 3) that connects each sample to both its primary neighbor and secondary connections, we establish a hierarchical clustering structure. This clustering mechanism naturally groups samples with analogous feature distributions.

Our analysis reveals an interesting architectural phenomenon: the distribution of clusters exhibits a clear pattern across network depth. Specifically, we observe a higher cluster density in the network's earlier layers compared to deeper ones. This observation suggests a functional specialization where shallow layers adapt to domain-specific representations, while deeper layers develop domain-invariant features that remain stable across different contexts.

---

**Algorithm 1** LFD Algorithm

---

1: **Input:** Feature map $F_j = \{1, 2, \cdots, B\}$ in the BN layer $j$, $F_j \in \mathbb{R}^{B \times C \times L}$, $1 \leq j \leq N$, where $N$ is total number of BN layers in the model.
2: **Output:** Feature map $F_j^{'} = \{F^1, F^2, \cdots, F^r\}$ after clustering, $F^i \in \mathbb{R}^{b \times C \times L}$, $1 \leq i \leq r$.
3: **Begin LFD Algorithm:**
4: Compute first neighbors integer vector $a^1 \in \mathbb{R}^{B \times 1}$ via exact distance (the distance metric is defined by Equation 2).
5: Given $a^1$, get adjacency matrix First of the feature map $F_j$ via Equation 3.
6: Generate connected components from matrix First.
7: **END**

---

# B    Details of Datasets

We evaluate adaptation performance under covariate shift using three benchmark datasets: CIFAR10-C, CIFAR100-C, and ImageNet-C [24]. These datasets simulate real-world distribution shifts through systematic perturbations at varying intensities, ranging from level 1 to 5, with higher levels representing more severe distribution shifts. The CIFAR-based corrupted datasets maintain their original class structures, with CIFAR10-C containing 10 categories (50K/10K split for train/test) and CIFAR100-C encompassing 100 categories (maintaining identical data volume). For large-scale evaluation, we utilize ImageNet-C, which spans 1,000 categories with approximately 1.28M training samples and 50K test instances. As shown in Figure 8, each dataset contains 15 types of corruptions, which are: Gaussian Noise, Shot Noise, Impulse Noise, Defocus Blur, Glass Blur, Motion Blur, Zoom Blur, Snow, Frost, Fog, Brightness, Contrast, Elastic Transformation, Pixelate, and JPEG Compression. These three datasets are sufficient to effectively reflect the performance of different methods in dynamic data streams and mixed domain scenarios.

Table 7: Comparison with state-of-the-art methods on ImageNet-R and ImageNet-$\bar{C}$ datasets (**batch size 64**). Results show **accuracy (%)** under CrossMix scenario using **ResNet-50**. Best results are shown in bold.

| Dataset | Source | $\alpha$-BN | IABN | TBN | TENT | EATA | SAR | DeYO | ViDA | RoTTA | NOTE | FIND |
|---|---|---|---|---|---|---|---|---|---|---|---|---|
| ImageNet-R | 27.94 | 31.60 | 21.63 | 31.12 | 31.36 | 31.65 | 31.22 | 32.05 | 31.12 | 30.78 | 21.20 | **34.74** |
| ImageNet-$\bar{C}$ | 33.78 | 38.01 | 23.35 | 38.63 | 38.69 | 39.00 | 37.90 | 38.42 | 38.49 | 39.62 | 24.48 | **41.91** |

To show the high extensibility of our method to other datasets. We compare our FIND with other methods on two additional datasets ImageNet-R [36] and ImageNet-$\bar{C}$ [37]. Based on the previous setup, we tested the performance of different methods under the CrossMix scenario (where all domains are mixed as input), with specific details as follows:

- **Style transfer-based dataset ImageNet-R:** ImageNet-R reflects style shift, where images are transformed from real photographs into various artistic styles (e.g., cartoons, sketches, etc.). Such shifts mainly affect low-level features such as edges and textures, resulting

in milder perturbations compared to typical corruptions. As shown in Table 7, our FIND achieves state-of-the-art performance, with improvements of 3%-13% over existing methods.

- **Human-made corruption dataset ImageNet-$\bar{C}$:** Compared to ImageNet-C, ImageNet-$\bar{C}$ includes a large variety of human-designed corruptions, such as Blue Noise, Plasma Noise, and Caustic Refraction. This allows for a more thorough evaluation of model robustness against structured, frequency-related, and complex synthetic corruptions. As shown in Table 7, our FIND still achieves the best results, outperforming the previous SOTA by 2.5%-17%.

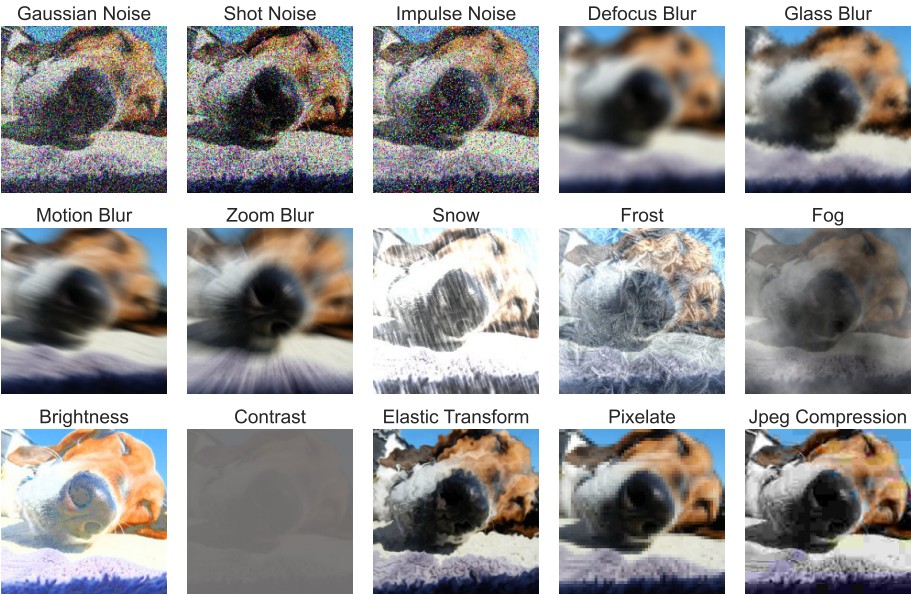

Figure 8: Visualization of the 15 types of corruptions in CIFAR10-C, CIFAR100-C and ImageNet-C.

## C  Details of Baselines

We consider the following baselines, including state-of-the-art test-time adaptation and test-time normalization algorithms.

**Test-time fine-tune.** Sharpness-aware and reliable entropy minimization method **(SAR)** [7] has the advantage of conducting selective entropy minimization, excluding samples with noisy gradients during online adaptation, which leads to more robust model updates. Additionally, SAR optimizes both entropy and the sharpness of the entropy surface simultaneously, ensuring the model's robustness to samples with remaining noisy gradients. Efficient anti-forgetting test-time adaptation **(EATA)** [21] improves the stability of model updates by filtering high-entropy samples, while applying Fisher regularizer to limit the extent of changes in important parameters, thereby alleviating catastrophic forgetting after long-term model adaptation. Destroy your object **(DeYo)** [8] disrupts the structural class-related features of samples by chunking them, and selects appropriate samples for model adaptation by comparing the entropy change in predictions before and after chunking, thereby enabling the model to learn the correct knowledge. Test entropy minimization **(TENT)** [4] optimizes the model for confidence as measured by the entropy of its predictions and estimates normalization statistics and optimizes channel-wise affine transformations to update online on each batch. Non-i.i.d. Test-time adaptation **(NOTE)** [3] is mainly two-fold: Instance-Aware Batch Normalization (IABN) that corrects normalization for out-of-distribution samples, and Prediction-balanced Reservoir Sampling (PBRS) that simulates i.i.d. data stream from non-i.i.d. stream in a class-balanced manner. Robust test-time Adaptation **(RoTTA)** [5] shares a similar approach with NOTE, which simulates an i.i.d. data stream by creating a sampling pool and adjusting the statistics of the batch normalization (BN) layer. Visual Domain Adapter **(ViDA)** [6] shares knowledge by partitioning high-rank and low-rank features. For the aforementioned methods that require updating the model, we follow

the online TTA setup. We assume that the source data, which is used for model pre-training, is not available for use in test-time adaptation (TTA). We conduct online adaptation and evaluation, continuously updating the model.

**Test-time normalization.** Test-time normalization. TBN [22] uses the mean and variance of the current input batch samples as the statistics for the BN layer. $\alpha$-BN [23] aggregates the statistics of TBN and SBN to obtain new statistics for the BN layer. IABN [3] is a method for calculating BN layer statistics in NOTE, which involves using the statistics of IN for correction. For these backward-free methods, we also follow the online TTA setting. Additionally, we do not make any adjustments to the model parameters. Instead, we only modify the statistics of the BN layer during the inference process using different approaches.

# D    Extension of Related Work

**Domain-aware multi-modeling in TTA:** CoLA [38] proposes a cross-device collaborative TTA paradigm, which achieves efficient TTA by sharing and aggregating domain knowledge vectors learned during adaptation across multiple devices. BECoTTA [39] aims to enhance the resistance to catastrophic forgetting and adaptation efficiency in the continual TTA process by introducing a Mixture-of-Experts (MoE) framework. DPCore [40] aims to address continual TTA under frequent and unpredictable target domain shifts, by preventing catastrophic forgetting and negative transfer during the adaptation process. UnMix-TNS [41] addresses non-i.i.d. scenarios with label imbalance by replicating K sets of BN statistics via Gaussian perturbations. However, these replicated statistics operate as an integrated whole, limiting their ability to capture diverse domain distributions in strict mixed-domain settings. These works differ significantly from ours in terms of motivation. Our work specifically targets the wild world setting with dynamic data streams-where the distribution diversity within each test batch changes arbitrarily-and the strict mixed-domain scenario. To address these issues, we propose a novel "divide-and-conquer" framework (automatic distribution-aware feature disentanglement).

**Robust test time fine-tuning in TTA:** COME [42] adopts subjective-logic–based uncertainty modeling and minimizes opinion entropy to address the overconfidence and instability (model collapse) caused by entropy minimization in test-time adaptation. REM [43] improves continual TTA stability and efficiency by preventing the model-collapse issues of entropy minimization. It introduces ranked entropy minimization: progressively masking object-relevant patches and jointly using masked consistency loss and entropy ranking loss to align predictions across difficulty levels while preserving the intended entropy order. EATA-C [44] reduces reducible model uncertainty via a full–subnetwork consistency loss and characterizes data uncertainty through prediction disagreements, thereby mitigating the overconfidence and miscalibration of pure entropy minimization in TTA while alleviating catastrophic forgetting. Although these methods improve stability during adaptation and prevent collapse, they still fail to achieve precise normalization in dynamic settings, preventing the model from learning effectively.

We compare against the above work that targets the same task as ours; the detailed results are shown in Table 8. Our FIND consistently outperforms these methods by a substantial margin (10%–15%).

Table 8: Comparison with state-of-the-art methods on CIFAR10-C, CIFAR100-C, and ImageNet-C datasets (**corruption severity 5, batch size 64**). Results show **accuracy (%)** under CrossMix scenario using **ResNet-50**. Best results are shown in bold.

| Dataset | COME | UnMix-TNS | REM | CoLA | FIND |
|---|---|---|---|---|---|
| CIFAR10-C | 60.48 | 60.46 | 59.10 | 62.00 | **71.54** |
| CIFAR100-C | 29.76 | 29.98 | 31.55 | 32.52 | **39.75** |
| ImageNet-C | 16.66 | 16.93 | 18.65 | 18.74 | **29.21** |

**The key differences between our work and prior "in-the-wild" work:** We consider fully dynamic data streams and more stringent domain mixing. Besides the CrossMix scenario—where each batch strictly includes 15 domains—we also construct dynamic scenarios (Random and Shuffle) where each batch may contain only a single domain or a varying number of domains (2–15). In addition, we provide a novel analytical perspective, redefining the roles of SBN and TBN in normalization from the viewpoint of domain-related and general features. We propose a divide-and-conquer normalization strategy for features within a batch—not addressed by previous methods. Notably,

DDA [10] introduces an additional diffusion model for denoising samples while keeping the original model unchanged. Although DDA also evaluates the mix domain scenario in its experiments, this is primarily to demonstrate the robustness of the method, rather than being the main focus of the paper. The differences between DDA and our work: 1. DDA operates on a per-sample basis. Even if a full batch is provided as input, each sample is denoised independently by the diffusion model. As a result, batch processing does not offer any additional algorithmic benefit beyond potential hardware-level parallelism. 2. DDA introduces additional diffusion models and requires multiple data forward passes, resulting in significant inference overhead. 3. Training the diffusion model requires large amounts of source data, which is inconsistent with TTA assumptions. In contrast, our method offers greater scalability across different batch sizes and achieves efficient inference without relying on additional models or extra training and forward processes.

## E  Details of Our Scenarios

Previous researches have predominantly focused on static scenarios (whether in continual TTA or non-i.i.d. TTA). As illustrated in Figure 9, the static scenario is characterized by samples within a batch originating from the same domain, with each domain persisting for an extended period.

In this paper, we shift our attention to TTA in dynamic scenarios. As depicted in Figure 9, the dynamic scenario is defined by samples within a batch originating from either a single domain or multiple domains, with domain changes occurring in real-time rather than persisting for extended periods. This encompasses three sub-scenarios:

- **CrossMix:** Each batch contains samples from multiple domains.
- **Shuffle:** Batches alternate between containing samples from a single domain and multiple domains.
- **Random:** The number of domains represented in each batch is entirely stochastic.

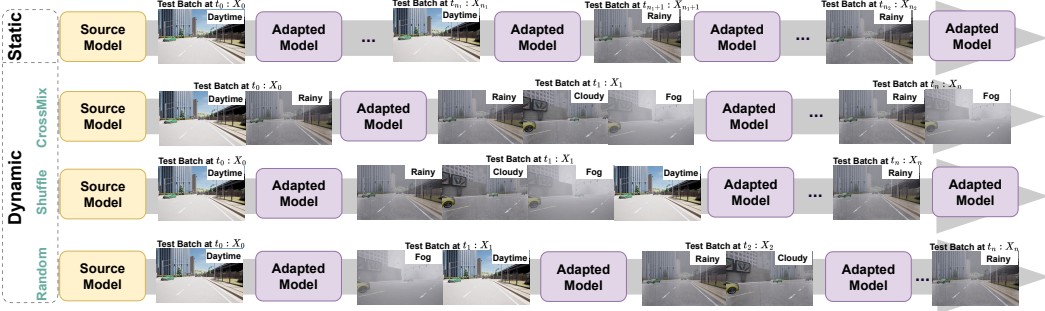

Figure 9: The visualization of scenarios mentioned in this paper, including static and dynamic (CrossMix, Shuffle and Random).

We propose three sub-scenarios that simulate real data streams in the wild world, while also encompassing continual TTA in the traditional TTA static scenario (where, under the Random scenario, the data stream may originate from the same domain over a certain period of time). Thus, these sub-scenarios provide a better evaluation of the performance of current TTA methods in real-world settings.

## F  Hyperparameter Settings

The hyperparameters can be divided into two types: one type is shared by all the baselines, and the other type consists of hyperparameters specific to each method. For the shared hyperparameters, $batch\ size = 64$, $learning\ rate = 1e^{-4}$. The optimizer used is SGD. The hyperparameters of each test-time fine-tuning method are set according to the TTAB benchmark [20] (the optimal hyperparameters that achieved the best performance for each method in the original paper). Following are the hyperparameters specific to each test-time normalization method:

- The hyperparameters of TBN are set according to the settings in [22];

- The hyperparameters of IABN are set according to the settings in [3];

- The hyperparameters of $\alpha$-BN are set according to the settings in [23].

## G  Cold-Start Mechanism and Layer Sensitivity

As shown in Figure 10a-Figure 10d, The sensitivity of different model layers to domain shift is correlated with both the backbone architecture and the dataset characteristics. For ResNet-50 (Figure 10a-Figure 10c), we observed an intriguing phenomenon: shallow and deep layers exhibit lower sensitivity to domain shift, while intermediate layers show heightened sensitivity. This deviates from conventional wisdom (where shallow layers are typically domain-related and deep layers class-related). The anomaly may arise because shallow layers predominantly capture universal features (e.g., object contours and edges) that are domain-invariant, whereas intermediate layers——transitioning from abstract to semantic representations——undergo intensive feature transformation and mixing, making them more domain-vulnerable. Furthermore, sensitivity correlates with task complexity, as evidenced by the uniformly elevated sensitivity across all layers in challenging tasks like ImageNet classification.

Figure 10e shows the sensitivity score values under different cold-start durations. We found that the cold-start duration has negligible impact on layer sensitivity quantification. Both short-term (3 batches) and long-term (100 batches) cold-start conditions consistently characterize the relative sensitivity relationships across layers, exhibiting only minor oscillations in peak values that do not compromise the final layer selection. We ultimately select the initial 10 batches as the cold-start duration.

**Overhead of cold-start phase:** The time overhead introduced by the cold-start phase is negligible. 1. The time complexity of the cold-start is linear with channels $C$ and the number of BN layers $N$, resulting in extremely low computational cost. 2. The computation is performed in parallel with the forward pass, incurring no extra inference or training steps.

## H  Analysis of Cluster Numbers of Each FABN Layer by LFD

Our analysis reveals the fundamental mechanism of LFD: it systematically organizes and clusters feature representations based on distributional similarities, thereby minimizing interference between dissimilar feature patterns within each batch. Figure 11 illustrates a key architectural phenomenon: the cluster count systematically decreases with network depth in dynamic scenarios. This pattern indicates a progressive convergence of feature distributions across deeper network layers, suggesting a transition from variable feature representations to more stable, consistent patterns. In Figure 11, each batch contains 15 domains and 10 classes. We observed an interesting phenomenon: in the shallow layers (0–5), the number of clusters slightly exceeds the number of domains; in the middle layers (6–40), the number of clusters

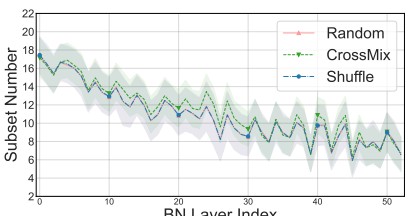

Figure 11: Cluster numbers of each FABN layer under dynamic (Random, CrossMix, and Shuffle) scenarios. Each result is derived from the mean of 1,000 batches, with the colored bands around the lines representing the standard deviation.

is almost identical to the number of domains; and in the deep layers, the number of clusters aligns closely with the number of classes. This aligns with the observations from the cold-start process: initially, the model focuses on general features such as edges and structures, which are more related to the objects themselves and less influenced by domains. As the layer depth increases, features transition from abstract to semantic, during which domains introduce interference. In the deep layers, features become directly tied to object categories, with minimal influence from domains. These findings provide valuable insights for optimizing the layer-wise scaling parameter $\alpha$ and understanding the functional roles of different network components.

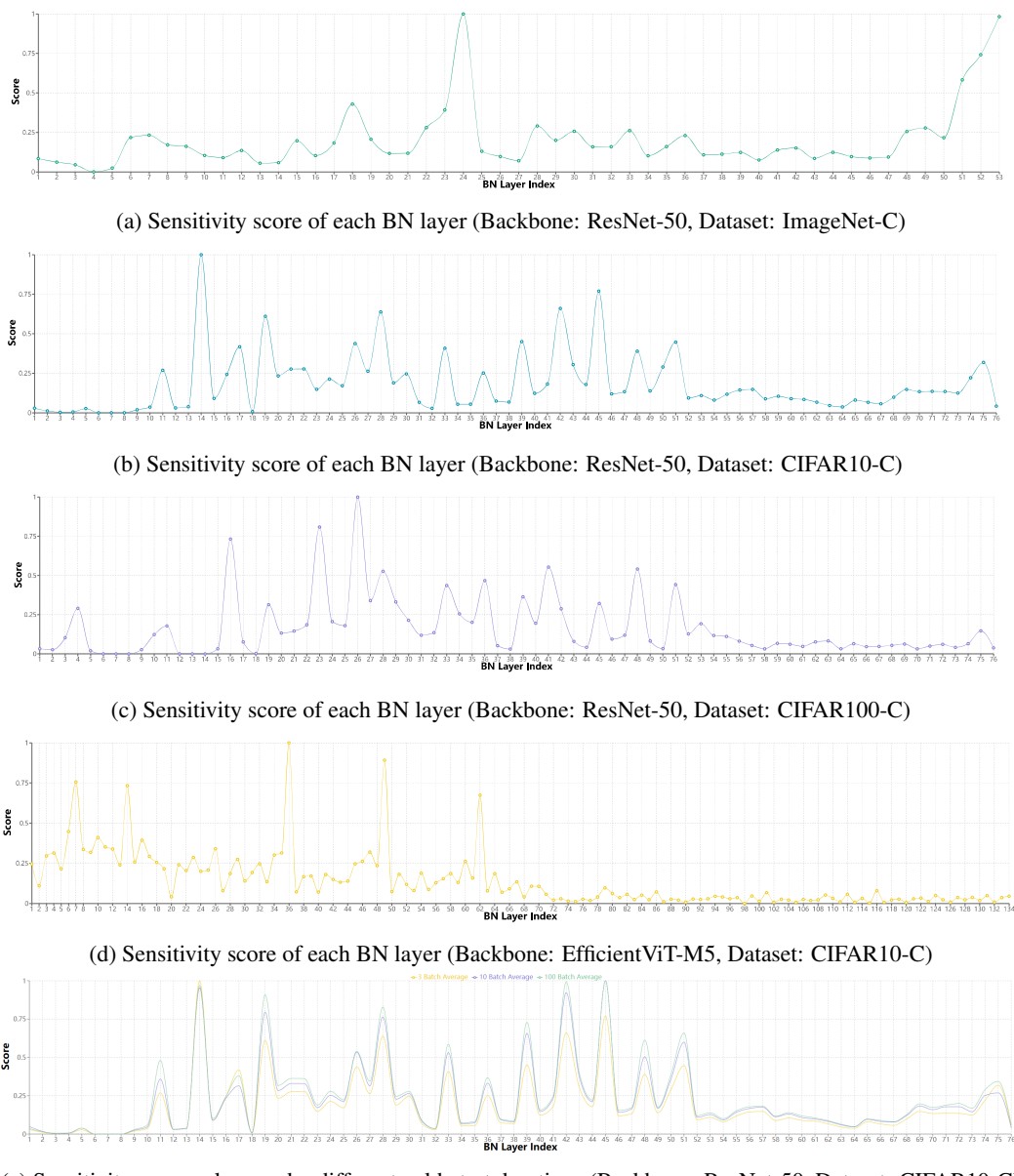

(a) Sensitivity score of each BN layer (Backbone: ResNet-50, Dataset: ImageNet-C)

(b) Sensitivity score of each BN layer (Backbone: ResNet-50, Dataset: CIFAR10-C)

(c) Sensitivity score of each BN layer (Backbone: ResNet-50, Dataset: CIFAR100-C)

(d) Sensitivity score of each BN layer (Backbone: EfficientViT-M5, Dataset: CIFAR10-C)

(e) Sensitivity score values under different cold-start durations (Backbone: ResNet-50, Dataset: CIFAR10-C)

Figure 10: (a)-(c) demonstrate the variation trends of domain shift sensitivity across BN layers under different backbones and datasets. (d) shows the sensitivity variations across BN layers in a single model under various cold-start durations.

# I  Sensitivity analysis of $\gamma$

We compress the score of each layer in FIND* to the range $[0, 1]$ through normalization, with score proximity to 0 indicating lower distribution shift sensitivity of the layer. As shown in Figure 12, when $\gamma$ varies within 0-0.1, accuracy fluctuation remains below 1% while inference efficiency improves by 41%-44%, demonstrating robust model performance. When $\gamma$ exceeds 0.1, accuracy drops by 2%-6% with unstable model behavior. Therefore, setting $\gamma$ between 0-0.1 achieves inference acceleration while preserving model performance. Meanwhile, the performance of the model is not sensitive to the $\gamma$ value within this interval.

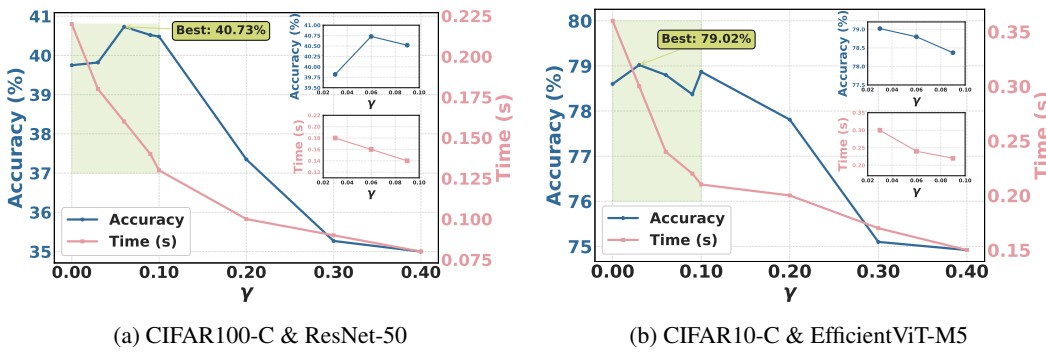

(a) CIFAR100-C & ResNet-50        (b) CIFAR10-C & EfficientViT-M5

Figure 12: (a) and (b) demonstrate the sensitivity of model inference performance (Accuracy (%)) to $\gamma$ variations, along with the associated changes in inference efficiency (Time (s)).

## J Compatibility Evaluation with Test-Time Tuning

We integrated FIND* with the test-time tuning methods discussed in our paper, and we froze the layers in S-FABN that require clustering while updating the layers that do not require clustering. As shown in Table 9, compared with FIND*, test-time tuning further improves performance by approximately 2%. We obtain the following conclusions: 1. Accurate normalization is a prerequisite for successful tuning (as stated in observation 1 of Section 2.2). 2. Selectively updating layers that are insensitive to domain shift yields greater gains.

Table 9: Performance of combining FIND* with test-time tuning on CIFAR10-C (**corruption severity 5, batch size 64**). Results show **accuracy (%)** under CrossMix scenario using **ResNet-50**. Best results are shown in bold.

|  | (a) FIND*+TENT [4] | | | | (b) FIND*+EATA [21] | | | | (c) FIND*+DeYO [8] | | | |
|---|---|---|---|---|---|---|---|---|---|---|---|---|
|  | $\alpha$=0.8 | $\alpha$=0.4 | $\alpha$=0 | $\alpha$=1 | $\alpha$=0.8 | $\alpha$=0.4 | $\alpha$=0 | $\alpha$=1 | $\alpha$=0.8 | $\alpha$=0.4 | $\alpha$=0 | $\alpha$=1 |
| $\gamma$=0.1 | **72.01** | 70.15 | 67.93 | 71.03 | **72.20** | 69.98 | 67.39 | 71.14 | **73.07** | 71.61 | 68.00 | 70.70 |
| $\gamma$=0.06 | **72.45** | 71.16 | 69.14 | 72.37 | **72.90** | 69.96 | 68.21 | 72.70 | **73.17** | 72.31 | 69.97 | 72.25 |

## K Performance under Different Domain Scales

Table 10 shows the performance of the method proposed in this paper under different domain scales in CrossMix Scenario. Experimental results demonstrate the superior adaptability of our approach across varying levels of distributional complexity in batch data. As the feature distributions become increasingly diverse (transitioning from homogeneous to heterogeneous batch compositions), our method maintains consistent performance advantages over existing approaches. Performance analysis reveals significant improvements: compared to the baseline methods, our approach achieves a 16% improvement over RoTTA and surpasses the previous state-of-the-art method DeYO by approximately 5%. Notably, across all three evaluation datasets, our framework consistently demonstrates a 2-5% performance gain over the strongest baseline. These results validate our method's effectiveness in handling dynamically changing domain distributions within batch data.

## L Performance under Different Model Structures

The generalization capabilities of our approach across different architectures are examined through ResNet-26 experiments, documented in Table 11. Under CrossMix evaluation conditions, our framework maintains its performance advantages regardless of the underlying network structure. Specifically, we achieve accuracy gains of approximately 3% over ViDA on CIFAR10-C and 4% over EATA on CIFAR100-C. These results demonstrate the architecture-agnostic nature of our method and its robust adaptation capabilities across varying model configurations.

Table 10: Comparisons with state-of-the-art methods on CIFAR10-C, CIFAR100-C, and ImageNet-C (severity level = 5) under **batch size = 64** regarding **accuracy (%)**. Each method was evaluated under the CrossMix scenario with various numbers of domains using a ResNet-50 model architecture. Best and second-best results are shown in bold and underlined, respectively.

| Method | CIFAR10-C | | | CIFAR100-C | | | ImageNet-C | | | Avg-All |
|---|---|---|---|---|---|---|---|---|---|---|
| | 6 Domains | 9 Domains | 12 Domains | 6 Domains | 9 Domains | 12 Domains | 6 Domains | 9 Domains | 12 Domains | |
| Source | 46.59 | 52.54 | 55.84 | 19.23 | 24.25 | 27.15 | 16.21 | 20.07 | 25.20 | 31.90 |
| **TEST-TIME FINE-TUNE** | | | | | | | | | | |
| TENT | 60.79 | 64.69 | 66.59 | 30.33 | 33.07 | 33.44 | 7.41 | 12.69 | 17.24 | 36.25 |
| EATA | 57.67 | 60.89 | 63.79 | **35.52** | 35.05 | 33.34 | 6.97 | 13.24 | 18.47 | 36.10 |
| NOTE | 58.61 | 61.88 | 64.93 | 28.55 | 31.46 | 33.65 | 12.92 | 17.23 | 22.71 | 36.88 |
| SAR | 59.76 | 63.76 | 65.75 | 31.03 | 33.48 | 33.71 | 7.44 | 12.75 | 18.05 | 36.19 |
| RoTTA | 52.03 | 35.44 | 42.26 | 23.44 | 21.73 | 23.72 | 6.96 | 16.16 | 22.37 | 27.12 |
| ViDA | 57.64 | 60.89 | 63.77 | 27.89 | 31.04 | 32.96 | 6.80 | 12.09 | 18.03 | 34.57 |
| DeYO | 65.23 | 69.22 | 69.89 | 35.35 | 35.71 | 33.96 | 7.39 | 13.59 | 17.85 | 38.69 |
| **TEST-TIME NORMALIZATION** | | | | | | | | | | |
| TBN | 57.63 | 60.89 | 63.77 | 27.89 | 31.03 | 32.95 | 7.39 | 12.39 | 17.51 | 34.60 |
| $\alpha$-BN | 58.17 | 61.68 | 64.27 | 28.49 | 31.79 | 33.81 | 9.78 | 15.36 | 21.80 | 36.13 |
| IABN | 55.30 | 59.41 | 64.24 | 18.99 | 21.99 | 25.62 | 3.55 | 6.44 | 10.30 | 29.54 |
| FIND | **66.59** | **70.30** | **72.97** | 35.08 | **38.82** | **40.68** | **17.05** | **22.38** | **27.40** | **43.47** |

Table 11: Comparisons with state-of-the-art methods on CIFAR10-C and CIFAR100-C respectively (severity level = 5) under **batch size = 64** regarding **accuracy (%)**. Each method was evaluated under the CrossMix scenario with using a ResNet-26 model architecture. The best result is denoted in bold black font.

| Method | CIFAR10-C | CIFAR100-C | Avg. |
|---|---|---|---|
| Source (ResNet-26) | 53.06 | 31.34 | 42.20 |
| **TEST-TIME FINE-TUNE** | | | |
| • TENT | 59.01 | 30.37 | 44.69 |
| • EATA | 59.34 | 34.24 | 46.79 |
| • NOTE | 60.18 | 31.28 | 45.73 |
| • SAR | 58.70 | 30.42 | 44.56 |
| • RoTTA | 49.26 | 20.10 | 34.68 |
| • ViDA | 65.80 | 30.52 | 48.16 |
| • DeYO | 65.47 | 33.89 | 49.68 |
| **TEST-TIME NORMALIZATION** | | | |
| • TBN | 59.34 | 30.52 | 44.93 |
| • $\alpha$-BN | 58.51 | 32.81 | 45.66 |
| • IABN | 62.31 | 19.72 | 41.02 |
| • FIND (ours) | **68.21** | **38.24** | **53.22** |

# M  Extension of Motivations

Figure 13 presents our analysis of model accuracy and TBN statistics across ImageNet-C corruption levels 1-5, where level 1 represents minimal domain shift. We evaluate the distributional alignment by measuring L2 norm and cosine similarity between instance normalization (IN) statistics and TBN statistics across batch normalization layers. Our results show that increasing corruption levels correlate with both declining accuracy and decreased statistical distance between instance-level and TBN features, suggesting stronger distributional coupling within batches.

These findings, illustrated through a representative example in Figure 14, demonstrate how elevated domain shifts compromise TBN's ability to maintain distinct class-specific feature distributions. The

observed pattern indicates that increasing distributional shifts in domain-related features interfere with TBN's capacity to characterize class-relevant features, leading to reduced discriminability across categories.

Based on these empirical observations and our theoretical analysis, we identify two fundamental limitations of TBN in dynamic scenarios: 1. Coupling between different domain-relevant features in TBN and 2. Interference between domain-related and class-relevant features in TBN. These findings underscore the importance of incorporating SBN into the adaptation framework.

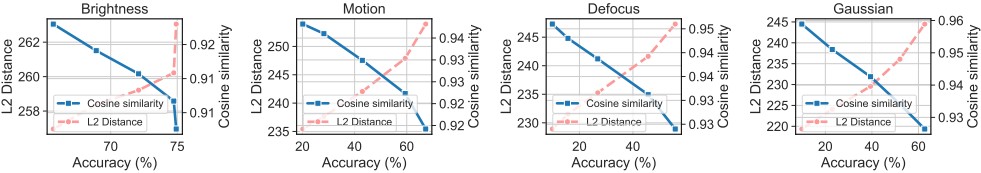

Figure 13: TBN-IN distance vs. accuracy under different corruptions. The 5 data points in the figure represent samples with corruption levels 1 through 5, where higher levels correspond to lower accuracy. We compute the average distance between per-sample IN statistics and TBN statistics in the deep layers, reflecting the dispersion of feature distributions within a batch. It reveals that the distribution of DRF interferes with the distribution of CRF: as the sample corruption level increases, the feature distributions within a batch become more coupled.

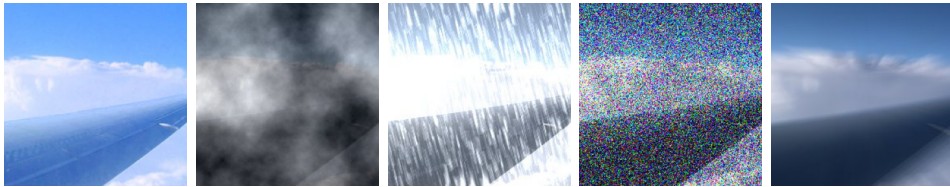

Figure 14: Airplane wings in different corruption. More severe corruption perturbations of the image can obscure the CRFs, causing the overall feature distribution to become more entangled and thus decreasing the model's inference accuracy.

## N   Performance under Static Scenario

Table 12 shows the results of the baselines and our method on ImageNet-C under the Static scenario. The data presented in the table demonstrates that both FIND and FIND* outperform the second-best method by approximately 3%. This result indicates that our approach maintains optimal performance in both dynamic and static scenarios.

Table 12: Comparisons with state-of-the-art methods on ImageNet-C (severity level = 5) under **batch size = 64** regarding **accuracy (%)**. Each method was evaluated under the static scenario using a ResNet-50 model architecture. Best and second-best results are shown in bold and underlined, respectively.

| Method | $\alpha$-BN | TBN | TENT | DeYO | RoTTA | SAR | EATA | ViDA | **FIND** | **FIND*** |
|---|---|---|---|---|---|---|---|---|---|---|
| ImageNet-C | 29.64 | 27.97 | 28.25 | 29.71 | 19.76 | 28.42 | 27.87 | 27.99 | 31.78 | **32.10** |

## O   Performance under Different Batch Size

The impact of varying batch sizes on adaptation performance under CrossMix conditions is illustrated in Figure 15. Our framework demonstrates remarkable stability across all evaluated batch size configurations on the three benchmark datasets, with minimal performance fluctuations. This contrasts sharply with alternative approaches, which exhibit significant sensitivity to batch size variations.

Notably, on CIFAR100-C and ImageNet-C, competing methods show substantial performance degradation with reduced batch sizes, only achieving stability at batch sizes exceeding 64.

Our method's batch-size invariance can be attributed to two key factors: the robust class-relevant feature representations provided by SBN, and the preservation of domain-specific characteristics regardless of batch size configurations. This architectural design enables consistent performance even in extreme scenarios - when processing individual samples (batch size = 1), the framework successfully captures domain-specific distributions from single-instance feature maps.

This stability across arbitrary batch sizes represents a significant advantage in practical deployment scenarios where batch size flexibility is crucial.

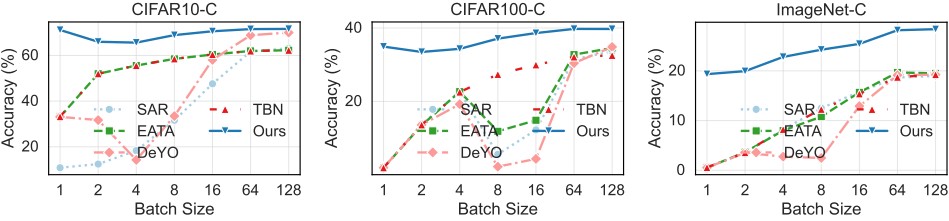

Figure 15: Batch size vs. accuracy. Other methods exhibit poorer performance at low batch sizes, only stabilizing as batch size increases. In contrast, our approach is insensitive to batch size variation, demonstrating greater robustness.

# P Experimental Results and Analysis under Wild Scenario

Table 13: Results for $\delta = 0.1$

| Method | 10-C | 100-C | Avg. |
|---|---|---|---|
| Source (ResNet-50) | 57.40 | 28.59 | 43.00 |
| **TEST-TIME FINE-TUNE** | | | |
| • TENT | 52.48 | 34.58 | 43.53 |
| • EATA | 54.99 | 35.60 | 45.30 |
| • NOTE | 67.50 | 24.70 | 46.10 |
| • SAR | 52.19 | 32.99 | 42.59 |
| • RoTTA | 49.49 | 22.60 | 36.05 |
| • ViDA | 55.02 | 36.72 | 45.87 |
| • DeYO | 50.77 | 18.27 | 34.52 |
| **TEST-TIME NORMALIZATION** | | | |
| • TBN | 55.00 | 36.73 | 45.87 |
| • $\alpha$-BN | 60.98 | **40.51** | 50.75 |
| • IABN | 63.30 | 24.92 | 44.11 |
| • FIND (ours) | **69.49** | 38.86 | **54.18** |

Table 14: Results for $\delta = 0.01$

| Method | 10-C | 100-C | Avg. |
|---|---|---|---|
| Source (ResNet-50) | 57.40 | 28.59 | 43.00 |
| **TEST-TIME FINE-TUNE** | | | |
| • TENT | 51.77 | 27.16 | 39.47 |
| • EATA | 54.47 | 24.39 | 39.43 |
| • NOTE | 67.49 | 24.71 | 46.10 |
| • SAR | 51.48 | 23.34 | 37.41 |
| • RoTTA | 49.27 | 22.33 | 35.80 |
| • ViDA | 54.47 | 30.59 | 42.53 |
| • DeYO | 50.35 | 11.27 | 30.81 |
| **TEST-TIME NORMALIZATION** | | | |
| • TBN | 54.47 | 30.57 | 42.52 |
| • $\alpha$-BN | 60.54 | 34.51 | 47.53 |
| • IABN | 63.28 | 24.97 | 44.13 |
| • FIND (ours) | **69.43** | **35.56** | **52.50** |

Table 15: Results for $\delta = 0.005$

| Method | 10-C | 100-C | Avg. |
|---|---|---|---|
| Source (ResNet-50) | 57.40 | 28.59 | 43.00 |
| **TEST-TIME FINE-TUNE** | | | |
| • TENT | 51.99 | 25.97 | 38.98 |
| • EATA | 54.42 | 23.42 | 38.92 |
| • NOTE | 67.56 | 24.70 | 46.13 |
| • SAR | 51.45 | 22.34 | 36.90 |
| • RoTTA | 49.14 | 21.87 | 35.51 |
| • ViDA | 54.42 | 29.81 | 42.12 |
| • DeYO | 48.04 | 10.17 | 29.11 |
| **TEST-TIME NORMALIZATION** | | | |
| • TBN | 54.42 | 29.80 | 42.11 |
| • $\alpha$-BN | 60.46 | 33.52 | 46.99 |
| • IABN | 63.30 | 24.95 | 44.12 |
| • FIND (ours) | **69.38** | **35.21** | **52.30** |

We extend the Random scenario by incorporating label distribution shifts to create the Wild scenario, which better approximates real-world data streams. This enhanced setup allows both distributional and label variations within batches, simulating open-world conditions. Following the methodology established in **NOTE** [3], we employ Dirichlet distributions to generate temporally correlated label sequences, with the concentration parameter $\delta$ ($\delta > 0$) controlling shift intensity - lower values indicating more severe shifts.

Results from experiments with varying shift intensities ($\delta = 0.1, 0.01, 0.005$) are presented in Tables 13, 14, and 15. Our framework demonstrates remarkable stability across increasing label shift intensities, maintaining consistent performance levels. Quantitatively, our approach achieves approximately 5% higher average accuracy compared to the strongest baseline, and a 20% improvement over the weakest performing method.

The performance gap becomes particularly pronounced under severe label shifts, where competing methods show significant degradation. For instance, DeYo's performance deteriorates substantially, dropping below 20% accuracy on CIFAR100-C under intense label shifts. This comparative analysis demonstrates our method's exceptional resilience to distribution variations in open-world scenarios, significantly outperforming existing approaches in handling complex, real-world data streams.

Table 16: Comparisons with state-of-the-art methods on CIFAR10-C and CIFAR100-C respectively (severity level = 5) under **batch size = 64** regarding **accuracy (%)**. Each method was evaluated under the CrossMix scenario ($Round = 2$) with using a ResNet-50 model architecture. Best and second-best results are shown in bold and underlined, respectively.

| Method | CIFAR10-C | CIFAR100-C | Avg. |
|---|---|---|---|
| Source (ResNet-50) | 57.41 | 28.59 | 43.00 |
| **TEST-TIME FINE-TUNE** | | | |
| • TENT | 58.89 | 27.20 | 43.05 |
| • EATA | 62.03 | 34.38 | 48.20 |
| • NOTE | 66.68 | 24.70 | 45.69 |
| • SAR | 57.54 | 27.59 | 42.56 |
| • RoTTA | 49.11 | 23.58 | 36.34 |
| • ViDA | 62.01 | 32.29 | 47.15 |
| • DeYO | 67.89 | 17.38 | 42.63 |
| **TEST-TIME NORMALIZATION** | | | |
| • TBN | 62.01 | 32.29 | 47.15 |
| • $\alpha$-BN | 62.45 | 33.21 | 47.83 |
| • IABN | 63.34 | 24.91 | 44.12 |
| • FIND (ours) | **71.48** | **39.72** | **55.60** |

Table 17: Comparisons with state-of-the-art methods on CIFAR10-C and CIFAR100-C respectively (severity level = 5) under **batch size = 64** regarding **accuracy (%)**. Each method was evaluated under the CrossMix scenario ($Round = 3$) with using a ResNet-50 model architecture. Best and second-best results are shown in bold and underlined, respectively.

| Method | CIFAR10-C | CIFAR100-C | Avg. |
|---|---|---|---|
| Source (ResNet-50) | 57.39 | 28.59 | 42.99 |
| **TEST-TIME FINE-TUNE** | | | |
| • TENT | 55.49 | 23.64 | 39.56 |
| • EATA | 62.01 | 33.67 | 47.84 |
| • NOTE | 67.90 | 24.70 | 46.30 |
| • SAR | 54.28 | 24.61 | 39.45 |
| • RoTTA | 49.21 | 22.11 | 35.66 |
| • ViDA | 62.02 | 32.21 | 47.12 |
| • DeYO | 65.77 | 12.77 | 39.27 |
| **TEST-TIME NORMALIZATION** | | | |
| • TBN | 62.02 | 32.21 | 47.12 |
| • $\alpha$-BN | 62.46 | 33.21 | 47.84 |
| • IABN | 63.28 | 24.87 | 44.08 |
| • FIND (ours) | **71.54** | **39.67** | **55.61** |

## Q    Experimental Results and Analysis on Simulated Lifelong Adaptation

While existing approaches focus on continuous adaptation to non-stationary data streams, their evaluation protocols typically utilize single-pass dataset sequences. To better approximate real-world deployment conditions, where inference cycles extend over longer periods, we propose an enhanced evaluation framework that extends test sequences through dataset replication. Specifically, we evaluate adaptation stability by creating extended sequences of CIFAR10-C and CIFAR100-C through two-fold and three-fold replication ($Round = 2$ and $Round = 3$). Results under CrossMix conditions are presented in Tables 16 and 17.

Our empirical analysis reveals distinct patterns in long-term adaptation stability. Our framework maintains consistent performance across extended sequences, showing minimal performance degradation between Round=2 and Round=3. In contrast, competing methods exhibit notable performance deterioration: DeYo experiences accuracy drops of 2% and 5% on CIFAR10-C and CIFAR100-C respectively. TENT shows performance degradation of 3% (CIFAR10-C) and 4% (CIFAR100-C). SAR demonstrates consistent 3% accuracy decline across both datasets. Other approaches show similar degradation patterns.

## R    License and Asset Attribution

We use the open-source implementations **TTAB** [20] under the Apache-2.0 license in our experiments.

TTAB is publicly available, and we have properly credited its creators in the main text. We have respected all terms of its license, and all usage is compliant with its respective open-source agreements.

## S    Broader Impacts

FIND improves the robustness of deep neural networks in dynamic real-world scenarios by addressing performance degradation under distribution shifts. This can enhance the reliability and adaptability of AI systems in diverse applications, including autonomous vehicles, medical diagnostics, and remote sensing, where data is often non-stationary and heterogeneous.

These improvements may contribute to more stable and efficient AI deployments, reducing the need for frequent model retraining and lowering computational costs. Additionally, the proposed framework's compatibility with both convolutional and transformer architectures broadens its accessibility for various use cases. We do not foresee negative societal impacts from this work.

## T    Limitations

While FIND significantly enhances model robustness in dynamic scenarios, the divide-and-conquer strategy used in Layer-Wise Feature Disentanglement (LFD) introduces additional computational overhead compared to simpler normalization techniques like TBN. Although our implementation is efficient and parallelized, the graph-based clustering in LFD may still become a bottleneck for extremely high-dimensional data or very large batch sizes. Further optimization of the clustering process and hardware-specific acceleration could address this issue.

## U    Conclusion and Future Work

In this work, we considered a more practical test scenario: The test samples within the same time period are no longer independent and identical, and may come from one or more different distributions. The existing TTA methods perform poorly in this scenario. Our analysis reveals that the main reason lies in the normalization failure of the BN layer. Therefore, we improved the existing test-time batch normalization methods and designed a new universal BN architecture. This architecture adopts a divide-and-conquer strategy and can precisely divide and normalize complex data streams during the testing period. In the experiment, we designed various scenarios, including: Dynamic, Static, Wild and Lifelong, to test the performance, robustness, practicability and stability under continuous learning of the method. We simultaneously adopt multiple backbones, including ResNet and ViT, to test the universality of the method. Experiments have proved that the method we designed has good effects in various scenarios during the test period and has strong practicability. Meanwhile, our method has achieved good results on multiple commonly used backbones by replacing the BN layer of the model, and has strong universality. Our work leaves several aspects unexplored. Specifically, we need to continue exploring the methods of online model updates so that we can also acquire effective knowledge when dealing with non-independent and non-distributed data streams. In addition, we also need to explore the optimal value of the FABN layer fusion parameter $\alpha$ to meet the specific characteristics of different layers.

