# OpenReview forum: "Feature-Based Instance Neighbor Discovery: Advanced Stable Test-Time Adaptation in Dynamic World"
_NeurIPS.cc/2025/Conference — NeurIPS 2025 poster_

### Official Review · Reviewer_dD3f · 2025-06-05

**Clarity:** 3
**Significance:** 2
**Originality:** 2
**Rating:** 5
**Confidence:** 4

**Summary:**

This paper proposes FIND, a method for dynamic test-time adaptation where test batches contain mixed distributions. FIND combines feature-based clustering (LFD), adaptive batch normalization (FABN), and selective application (S-FABN) to improve robustness under distribution shifts. Experiments show gains in accuracy and efficiency compared to existing TTA methods.

**Questions:**

1. Comparison with DDA: The proposed setting seems closely related to DDA. Could the authors provide a direct experimental comparison with DDA under the same evaluation protocol? Additionally, a more detailed discussion distinguishing the problem settings and assumptions would help clarify the novelty of this work.
2. Generality Across Architectures: The current experiments are limited to ResNet-50. Have the authors tested the proposed method on other network architectures such as ViTs or ConvNeXt? Demonstrating consistent performance improvements across diverse model families would enhance the method’s perceived robustness and practical utility.

**Ethical Concerns:**

["NO or VERY MINOR ethics concerns only"]

**Final Justification:**

My recommendation is to Accept.

I thank the authors for their detailed rebuttal. The responses have successfully addressed the main concerns and questions I raised in my initial review.

Resolved issues: differences between DDA and the work, generality across architectures

Unresolved issues: none

**Limitations:**

yes

**Quality:**

3

**Strengths And Weaknesses:**

## Strengths
1. The paper is well written, with clear explanations that make the technical content accessible. The logical flow and precise language enhance the overall readability.
2. The visualizations, including figures and tables, are carefully designed and effectively support the narrative. They provide clear and intuitive illustrations of the proposed method and its empirical results.


## Weaknesses
1. The problem setting considered in this work appears closely related to DDA [1]. While DDA is cited in the introduction, the paper does not provide a direct empirical comparison against DDA, nor does it offer a thorough discussion clarifying the differences between the two settings. This omission makes it difficult to precisely assess the novelty and contributions relative to existing work.
2. The experimental validation is limited to ResNet-50. The lack of evaluation on a broader range of network architectures raises concerns regarding the generalizability and robustness of the proposed method across different model backbones.

[1] Gao J, Zhang J, Liu X, et al. Back to the source: Diffusion-driven adaptation to test-time corruption[C]//Proceedings of the IEEE/CVF Conference on Computer Vision and Pattern Recognition. 2023: 11786-11796.

---

> ### Author Rebuttal · Authors · 2025-07-27
>
> ## Thank you for recognizing the well-organized structure and clear logic of our paper, as well as your acknowledgment of the supportive illustrations and empirical results. Your suggestions are helpful for us to further clarify our core contributions. Our response to your concerns is presented as follows.
>
> >## **Q1 (Clarify differences between DDA and our work)**
>
> We are extremely grateful to the reviewer for this valuable suggestion, which will help us further distinguish our work from others.
>
> **There are significant differences between our FIND  and DDA [1]. We have sufficient innovation in motivation, methods and analysis.** We will describe in detail below.
>
> **DDA focuses on shifting from model adaptation to input data adaptation, with the core motivation of avoiding frequent updates to the model itself during TTA and instead directly adapting the input data to prevent model collapse**. To this end, DDA introduces an additional diffusion model for denoising samples while keeping the original model unchanged. Although DDA also evaluates the mix domain scenario in its experiments, this is primarily to demonstrate the robustness of the method, rather than being the main focus of the paper.
>
> In contrast, our motivation lies in **addressing dynamic data streams and strict mix domain scenarios in the wild world**, which are fundamentally different from the static settings commonly assumed in traditional TTA. Our goal is to enhance inference performance and robustness in real-world environments. To this end, we construct a variety of dynamic scenarios simulating data distributions in the wild, including 1. strict mix domain, where each input batch contains samples from multiple different domains (up to 15), and 2. dynamically changing distributions, where the distribution diversity in each batch varies randomly (with the number of domains ranging from 1 to 15). To address these challenges, we propose a novel "divide-and-conquer“ normalization framework. Unlike traditional normalization methods that process the entire batch as a whole, our approach can automatically perceive distribution diversity within a batch and achieve layer-wise feature disentanglement.
>
> Furthermore, we examine different normalization schemes from the perspective of both general features and domain-related features, leading to several novel conclusions (see Section 2.2 and Appendix K) that are not provided by DDA.
>
> Other differences of DDA compared with us:  1. DDA operates on a per-sample basis: even if a full batch is provided as input, each sample is denoised independently by the diffusion model. As a result, batch processing does not offer any additional algorithmic benefit beyond potential hardware-level parallelism. 2. DDA introduces additional diffusion models and requires multiple data forward passes, resulting in significant inference overhead.  3. Training the diffusion model requires large amounts of source data, which is inconsistent with TTA assumptions.  In contrast, our method offers greater scalability across different batch sizes and achieves efficient inference without relying on additional models or extra training and forward processes.
>
> As shown in the table below, our method outperforms DDA, achieves **high efficiency (requiring only 1% of DDA's time cost)**. We will provide additional explanation of the differences between DDA and our method, and include these results in the revised version. We thank the reviewer for raising this important point.
>
> |        | FIND         | DDA            |
> |:------:|:------------:|:--------------:|
> | Acc/%  | 30.33        | 26.10          |
> | Efficiency | 0.15s/1.21GB | 19.83s/12.88GB |
>
> [1]  Back to the source: Diffusion-driven adaptation to test-time corruption. CVPR 2023.
>
> >## **Q2 (Generality Across Architectures)**
>
> Thank you for raising this important question!
>
> **We have provided evaluations on different architectures in the paper.** In our experiments, we provide two general architectures: CNN-based models (ResNet-50, ResNet-26) and  ViT-based models (Efficient-ViT). The results are reported in Tab. 1 and Tab. 2 of the main paper, as well as Tab. 7 in the appendix. Our method demonstrates strong generalizability and plug-and-play flexibility on different architectures.
>
> Thanks again for your valuable feedback！We will emphasize these tests in the revised version.

---

> > ### Comment · Reviewer_dD3f · 2025-08-01
> > **Response to Rebuttal**
> >
> > I thank the authors for their detailed rebuttal. The responses have successfully addressed the main concerns and questions I raised in my initial review.

---

> > > ### Author Response · Authors · 2025-08-01
> > >
> > > Thanks for your valuable time and quick reply. We appreciate your suggestions and will add these additional results to the revision. Please let us know if you need more information. We greatly appreciate this opportunity to improve our work. Could you please reconsider  your rating to reflect our efforts in the rebuttal?

---

### Official Review · Reviewer_4ioQ · 2025-06-30

**Clarity:** 3
**Significance:** 3
**Originality:** 3
**Rating:** 5
**Confidence:** 5

**Summary:**

The paper proposes FIND, a novel test-time adaptation method designed for dynamic scenarios where test samples may come from multiple distributions. FIND addresses the limitations of existing normalization strategies by introducing a divide-and-conquer approach, comprising Layer-wise Feature Disentanglement (LFD) to partition features, Feature Aware Batch Normalization (FABN) to combine source and test-specific statistics, and Selective FABN (S-FABN) to enhance efficiency. Extensive experiments on benchmark datasets demonstrate FIND's superior performance, achieving significant accuracy improvements over state-of-the-art methods while maintaining computational efficiency.

**Questions:**

Please refer to the Weaknesses.

**Ethical Concerns:**

["NO or VERY MINOR ethics concerns only"]

**Final Justification:**

The authors' rebuttal has addressed my concerns, and I will maintain my positive scores.

Resolved issues: More experiments about different datasets, TTA settings, Efficiency

Unresolved issues: none

**Quality:**

3

**Strengths And Weaknesses:**

Strengths:
1. The paper introduces a novel divide-and-conquer normalization approach tailored for dynamic scenarios, effectively addressing the issue of interference between different distributions and leading to a 30% accuracy improvement in dynamic scenarios compared to existing methods.
2. FIND demonstrates robust performance across various architectures and datasets, achieving superior results on both convolutional networks and transformer-based models while maintaining consistent accuracy regardless of batch size.
3. FIND achieves significant computational efficiency with low memory usage and fast inference speed, making it highly practical for real-world deployment.

Weaknesses:
1. The experiments are conducted on a limited set of benchmark datasets (CIFAR10-C, CIFAR100-C, and ImageNet-C). While these datasets are widely used, they may not fully capture the diversity of real-world distribution shifts. Evaluating FIND on a broader range of datasets could provide a more comprehensive assessment of its robustness and effectiveness.
2. The cold-start phase, which involves computing the average sensitivity score for each layer, may introduce additional computational overhead at the beginning of the adaptation process. This could be a drawback in scenarios where rapid adaptation is required.
3. How does the proposed method perform on larger models with high-dimensional feature spaces? Whether the computational complexity of the layer-wise feature disentanglement and selective normalization becomes prohibitive?
4. More related methods should be discussed or compared in the manuscript for a comprehensive analysis, including but not limited to [A-D].

Minor issues:
1. In Eqn.(1), \sum_L  F_{i;c;L} should be \sum_{l=0}^L F_{i;c;l}.
2. In line 193, F_{j;c;L} is not mentioned in any equation.

References:
[A] Ranked Entropy Minimization for Continual Test-Time Adaptation, ICML 2025.
[B] Uncertainty-Calibrated Test-Time Model Adaptation without Forgetting, TPAMI 2025.
[C] Test-Time Model Adaptation for Visual Question Answering With Debiased Self-Supervisions, TMM 2023.
[D] COME: Test-time adaption by Conservatively Minimizing Entropy, ICLR 2025.

---

> ### Author Rebuttal · Authors · 2025-07-27
>
> ## We sincerely appreciate your recognition of the novelty, effectiveness, and robustness of our method, as well as its high efficiency and practical value. We greatly value your constructive feedback. Our response to your concerns is as follows.
>
> >## **Q1 (Use more dataset to evaluate)**
>
> We sincerely appreciate the reviewer’s suggestion, which further demonstrates the effectiveness of our approach.
>
> We adopted evaluation benchmarks from several classic TTA methods (e.g., TENT) as well as wild world TTA methods (e.g., SAR), namely CIFAR10-C, CIFAR100-C, and ImageNet-C. These three datasets are sufficient to effectively reflect the performance of different methods in dynamic data streams and mixed domain scenarios. **High extensibility to other datasets**: Our method is also eadily extensible to other datasets, and we provide two examples here.  We will provide the complete test results on as many datasets as possible in the revised version.
>
> - Style transfer-based dataset ImageNet-R [1]: ImageNet-R reflects **style shift**, where images are transformed from real photographs into various artistic styles (e.g., cartoons, sketches, etc.). Such shifts mainly affect low-level features such as edges and textures, resulting in milder perturbations compared to typical corruptions. Our FIND method  **achieves state-of-the-art performance**, with improvements of **3%-13%** over existing methods.
>
> - Human-made corruption dataset ImageNet-$\overline{C}$ [2]:  Compared to ImageNet-C, ImageNet-$\overline{C}$ includes a large variety of human-designed corruptions, such as **Blue Noise, Plasma Noise, and Caustic Refraction**. This allows for a more thorough evaluation of model robustness against structured, frequency-related, and complex synthetic corruptions. Our FIND method still **achieves the best results**, outperforming the previous SOTA by **2.5%-17%**.
>
> | Method            | Source | $\alpha$-BN | IABN  | TBN   | TENT  | EATA  | SAR   | DeYO  | ViDA  | RoTTA | NOTE  | FIND (Ours) |
> |-------------------|---------------|-------------|-------|-------|-------|-------|-------|-------|-------|-------|-------|-------------|
> | **ImageNet-R**                | 27.94         | 31.60      | 21.63 | 31.12 | 31.36 | 31.65 | 31.22 | 32.05 | 31.12 | 30.78 | 21.20 | 34.74       |
> | **ImageNet-$\overline{C}$**   | 33.78         | 38.01      | 23.35 | 38.63 | 38.69 | 39.00 | 37.90 | 38.42 | 38.49 | 39.62 | 24.48 | 41.91       |
>
> [1] The many faces of robustness: A critical analysis of out-of-distribution generalization. ICCV 2021.
>
> [2] On interaction between augmentations and corruptions in natural corruption robustness. NeurIPS 2021.
>
> We will include evaluation results on these datasets in the revised version.  We would like to express our sincere gratitude once again for the valuable suggestions from the reviewer.
>
> Extended explanation：CIFAR10-C, CIFAR100-C, and ImageNet-C contain 15 types of common real-world corruptions, which can be mapped to four phenomena frequently encountered in practice: **noise (affecting the entire image at the pixel level), weather (affecting local or global areas), digital artifacts (affecting pixel blocks, color areas, or the whole image), and blur (affecting edges and details)**. These are caused by real-world factors such as signal interference, changing weather conditions, image compression, motion blur, or defocus during photography, making them highly representative of wild-world scenarios. Such corruptions impact the overall visual quality and introduce domain shift. In our paper, we follow prior works exploring wild-world settings (e.g., SAR) to evaluate our method on these three datasets. We find that these datasets are **particularly challenging** in the dynamic and CrossMix scenario, as corruptions affect the entire image in diverse ways, further intensifying covariate shift. Therefore, to best demonstrate the realism and practical significance of our scenario, we adopt these three datasets.
>
>
> >## **Q2 (Overhead of cold-start)**
>
> We believe the question you raised is highly valuable. We will add the results in the revised version. Thanks for your good suggestions.
>
>  **The time overhead introduced by the cold-start (CS) phase is negligible.** 1. The time complexity of the cold-start  is linear with channels $C$ and the number of BN layers $N$, i.e., $O(CN)$, resulting in extremely low computational cost. 2. The computation is performed in parallel with the forward pass, incurring no extra inference or training steps. The following table presents the test results with and without the cold-start phase:
>
> | Model      | With CS   | Without CS |
> |:----------:|:---------:|:----------:|
> | ResNet-50  | 0.20728 s | 0.20728 s  |
>
> >## **Q3 (Efficiency of FIND)**
>
> Thanks for raising this important question.
>
>  **FIND remains highly efficient even in high-dimensional models.**
>
> - Time Complexity: The main overhead comes from the first neighbor search, which is $O(B^2 n C)$, where $B$ is the batch size, $n$ is the number of clustering layers ($n \ll N$, $N$ is the total number of BN layers), and $C$ is the number of channels. With nearest-neighbor algorithm optimization, this can be further reduced to $O(B \log B n C)$.
>
> - Conclusion: **The time cost scales linearly with $C$ and $n$.**
>
> - Empirical Results: When the number of BN layers is increased by **2.5$\times$ (From 53 to 135)**, the inference time increases by only **50% (From  0.16s to 0.24s)**. For further improvement, we can set $\gamma$ to 0.2, which further increases inference efficiency **(20% improvement)** while incurring negligible performance loss (around 1%).
>
> | Efficient-ViT           | Time/s | ResNet-50           | Time/s |
> |:-----------------------:|:------:|:-------------------:|:------:|
> | All Layers 135/135      | 0.41   | All Layers 53/53    | 0.21   |
> | $\gamma$ = 0.1  (60/135)   | 0.24   | $\gamma$ = 0.1  (38/53)| 0.16   |
> | $\gamma$ = 0.2  (38/135)   | 0.20   | $\gamma$ = 0.2  (17/53)| 0.13   |
>
> >## **Q4 (Compare FIND with more methods)**
>
> Thanks for this good suggestions. We will include a discussion of these methods in the revised version and conduct comparative experiments.
>
> Our initial selection of baselines encompassed a diverse and extensive collection of SOTA methods, with a total of 10 approaches included. To address your concerns, we have identified the following works as relevant to our discussion: COME (ICLR 2025)[1], UnMix-TNS (ICLR 2024) [2], DDA (CVPR 2023)|[3], REM (ICML 2025) [4], CoLA (NIPS 2024)  [5].
>
> The results are shown in the table below. Our FIND consistently surpasses these methods by a large margin **(10%-15%)**. This further supports our conclusion that precise normalization plays a dominant role in effective TTA (Section 2.2, Conclusion 2).
>
> Note: The original DDA paper only conducts experiments on IN-C and does not provide diffusion models for other datasets.
>
> Acc (%) of CrossMix, ResNet-50
> |           | COME [1] | UnMix-TNS [2] | DDA [3] | REM [4] | CoLA [5]  | FIND (Ours) |
> |-----------|----------|---------------|---------|---------|----------|-------------|
> | 10-C      | 60.48%   | 60.46%        |   -       | 59.10%  | 62.00%     |71.54%      |
> | 100-C     | 29.76%   | 29.98%        |  -       | 31.55%  | 32.52%    |40.48%      |
> | IN-C      | 16.66%   | 16.93%        | 26.10%  | 18.645% | 18.74%   |30.33%      |
>
> FIND is an efficient and effective method for handling dynamic and mixed data streams. Our "divide-and-conquer" normalization strategy is novel and enables FIND to effectively capture distribution diversity and perform feature partitioning.
> Thanks again for your valuable suggestions. We will add the descriptions and complete results of these newly given methods in the revised version. Some methods are not compatible with our tasks or framework, or have not released their code. We will also refer to and explain the differences in revised version.
>
> [1] COME: Test-time adaption by Conservatively Minimizing Entropy. ICLR 2025.
>
> [2] Un-Mixing Test-Time Normalization Statistics: Combatting Label Temporal Correlation. ICLR 2024.
>
> [3]  Back to the source: Diffusion-driven adaptation to test-time corruption. CVPR 2023
>
> [4] Ranked Entropy Minimization for Continual Test-Time Adaptation, ICML 2025.
>
> [5]  Cross-Device Collaborative Test-Time Adaptation. NIPS 2024.

---

> > ### Comment · Reviewer_4ioQ · 2025-08-02
> > **Response to Rebuttal**
> >
> > I appreciate the authors’ detailed responses. They have adequately addressed my concern, and I will retain my positive score.

---

> > > ### Author Response · Authors · 2025-08-02
> > >
> > > We greatly appreciate the reviewer’s positive response to our revision and will further polish the manuscript accordingly. Once again, we sincerely thank you for your recognition of our work.

---

### Official Review · Reviewer_wnbc · 2025-06-30

**Clarity:** 3
**Significance:** 3
**Originality:** 3
**Rating:** 4
**Confidence:** 4

**Summary:**

In this paper, authors consider test-time adaptation under a more complicated real-world scenario, where multiple distinct distributions may emerge simultaneously. This impose new challenges to previous test-time adaptation methods. Motivated by this, authors propose to  exploit the per-lay feature map similarity among test samples in the same mini-batch to construct graph structures. Afterwards, batch normalization is applied under the combination of source batch normalization (SBN) statistics and test-feature-specific-normalization (TFN) statistics, which are computed on a per graph connected component basis. This captures domain-specific feature distributions and also alleviates statistical bias in TFN estimations with the introduction of SBN statistics. In addition, to improve the computation efficiency, authors propose to dynamically determine which layer should be handled by graph partition, thus reducing unnecessary computation for layers insensitive to domain shifts. Empirical evaluation on ResNet and ViT models further proves the effectiveness of the proposed method.

**Questions:**

1. Section 2.4, Eq. (5) and Eq. (6). Based on my understanding, the feature aware batch normalization (FABN) is proposed to resolve the potential statistical bias in TFN estimations when the corresponding subset is small. Thus, the relative contribution of source batch normalization (SBN) should be dependent on the subset size, rather than fixed as a hyper-parameter $\alpha$.
2. Section 2.5, line 238. The selective feature aware batch normalization (S-FABN) relies on a cold-start phase to esimate the average score for each layer. What if the test distribution changes significantly after the cold-start phase? Will this leads to performance degradation?
3. Sensitivity analysis of $\alpha$, the trade-off coefficient between SBN and TFN, is missing.
4. Section 3.6, Table 4. $\alpha$-BN is not included in the comparison of memory and latency of different test-time adaptation methods.
5. Section 3.7, Figure 7. The proposed method works well even under small batch size, which is quite interesting. For instance, with BS=1, the graph partition should have no effect completely, then why the proposed method still outperforms all the baseline methods?

**Ethical Concerns:**

["NO or VERY MINOR ethics concerns only"]

**Final Justification:**

Authors have addressed my concerns in the rebuttal. Therefore, I will raise my rating accordingly.

**Limitations:**

Yes, limitations are provided in the appendix.

**Paper Formatting Concerns:**

None.

**Quality:**

3

**Strengths And Weaknesses:**

Strengths:
* The test-time adaptation scenarios with multiple test distributions considered in the paper is well motivated and more closely-related to real-world settings. Authors identify the performance degradation of existing time-time adaptation methods under this setting, which naturally leads to the proposal of layer-wise feature disentanglement.
* Authors also point out that not all the layers should be handled with similarity-based graph construction and feature-aware batch normalization. Instead, they propose to use the KL divergence to measure the shift between source and current target domains, which serves as the criterion for whether graph partition is needed for the current layer. This also leads to more efficient inference.

Weaknesses:
* Althought the selective feature aware batch normalization (S-FABN) is well motivated, the design of KL-divergence based criterion seems somehow problematic. Since graph partition is needed when there are multiple test distributions presented in a single mini-batch, then the criterion should reflect the distribution diversity among test samples, rather the difference between source and target domains.
* Some experimental results are not thoroughly analyzed (please refer to subsequent questions for details). Besides, sensitivity analysis of $\alpha$ and efficiency comparison against $\alpha$-BN is missing.

---

> ### Author Rebuttal · Authors · 2025-07-28
>
> ## Thank you for acknowledging the motivation behind our work, the valuable analysis and insights, and the high efficiency of our methods. Your constructive feedback and suggestions provide great inspiration for our future exploration. Our response to your concerns is presented as follows.
>
> >## **W1 (The criterion of S-FABN)**
>
> Thanks for  raising this important question!
>
> **We would like to clarify that the criterion in S-FABN is designed to measure intrinsic properties of the model itself and is independent of the input data**. Specifically, the criterion measures the discrepancy between source and target feature distributions at each layer, allowing us to assess the layer’s sensitivity to domain shift. A large discrepancy indicates that the layer is domain-sensitive, i.e., it focuses on domain-related features, and thus distribution-aware partitioning is required. In contrast, a small discrepancy indicates that the layer is domain-insensitive, i.e., it mainly captures generic features shared across domains, so partitioning can be skipped to improve **inference efficiency**.
>
> Our LFD (Section 2.3 in our paper) module already possesses automatic distribution diversity awareness. What we need now is a method to further improve efficiency (S-FABN). Our method is motivated by the challenge of handling data streams with various distribution diversity, for which we propose LFD—a mechanism that can automatically sense distribution diversity and perform feature partitioning. S-FABN is specifically designed to address the following issues encountered in LFD: 1. Do all BN layers in the model require LFD for distribution-aware feature partitioning? 2. Are there any layers that are insensitive to domain-related features (i.e., not affected by domain shift), such that the features in this layer are shared across both source and target domains (generic features)? If such layers exist, LFD is unnecessary for them and batch statistics can be computed directly over the entire batch, significantly improving efficiency.  **Therefore, we propose the criterion based on KL divergence in S-FABN to identify which layers focus on features that are similar between the source and target domains, thereby avoiding the additional overhead brought by LFD.**
>
> Experimental results: We evaluated on batches containing various numbers of  domains (Tab. 6 in the Appendix), as well as scenarios where the number of domains per batch varies randomly from 1 to 15 (Random and Shuffle in Tab. 1 & 2). Our LFD effectively **senses and partitions features** according to the degree of distribution diversity, achieving optimal performance.  Therefore, we do not need to continue designing indicators that can reflect distribution diversity.
>
> Conclusion:  S-FABN identifies layers that are **insensitive to domain shift**, allowing us to skip LFD  for these layers and further improve efficiency.  Thank you again for raising this very valuable question. It touches on a key point of our paper.
>
>
> >## **Q1 (why $\alpha$ is fixed)**
>
> We thank the reviewer for this insightful question!
>
>  **The setting of $\alpha$ is almost imperceptible to the subset size.** SBN provides a comprehensive and robust distribution of generic knowledge (Section 2.2, line 164 in our paper), which is  clearer and more stable compared to TFN/TBN (Appendix K). Therefore, **SBN should be dominant in the combination**, while TBN/TFN mainly supply the  domain-related feature distribution. This can be simply formulated as:
>
> - **generic knowledge (SBN) + domain-related distribution (TFN/TBN) constructs complete distribution**.
>
> - Experimental results: We fix the $\alpha$ of un-clustered layers to 0.8 and varying $\alpha$ in clustered layers from [0, 0.2, 0.4, 0.6, 0.8, 1]. Under both **batch size = 64 and batch size = 512**, we consistently observe better performance when SBN dominates ($\alpha > 0.5$). Notably, when $\alpha = 1$, performance drops sharply, indicating that without any contribution from TFN, i.e., no domain-related distribution, the results are suboptimal.
>
> **BS = 64**, $\alpha =0.8$ for Non-Clustered Layers (TBN+SBN) and changes in [0, 0.2, 0.4, 0.6, 0.8, 1] for Clustered Layers (TFN+SBN). $\alpha$ represents for the weight of SBN
> | $\alpha$ | 0    | 0.2  | 0.4  | 0.6  | 0.8  | 1    |
> |:--------:|:----:|:----:|:----:|:----:|:----:|:----:|
> | Acc/%    |57.48 |62.98 |66.81 |69.49 |70.75 |60.39 |
>
> **BS = 512**, $\alpha =0.8$ for Non-Clustered Layers (TBN+SBN) and changes in [0, 0.2, 0.4, 0.6, 0.8, 1] for Clustered Layers (TFN+SBN). $\alpha$ represents for the weight of SBN
> | $\alpha$ | 0    | 0.2  | 0.4  | 0.6  | 0.8  | 1    |
> |:--------:|:----:|:----:|:----:|:----:|:----:|:----:|
> | Acc/%    | 59.28| 63.28| 66.90| 69.50| 71.15| 60.34|
>
> Conclusion: **SBN needs to be dominant**, as it provides comprehensive and stable generic knowledge,  which is also **the primary reference during inference**. While TFN can offer relatively stable generic knowledge when the batch size is extremely large, in practice:
>
> - Batch size rarely exceeds 1e3. Even with a larger batch size, the statistics from TFN are still less comprehensive than those from SBN, which are computed over tens of thousands of training samples.
>
> Thanks again for your excellent question! We will provide a detailed explanation of the choice of $\alpha$ in the revised version.
>
> >## **Q2 (Is S-FABN affected by data distribution?)**
>
> Thanks for raising this insightful question!
>
>  **There is no performance degradation**. The Score measures the **intrinsic properties of different layers in the model** and is largely **independent of input data distribution changes**. Our tests have shown that certain layers are insensitive to domain shift—they focus mainly on generic features. For these layers, there is no need to perform distribution sensing or partitioning on the input batch; changes in the domain of input samples do not affect them, as these layers attend to the **invariant, generic features**.
>
> Test results:  Fig. 10e in  appendix demonstrates this point: we measured the average Score after **3, 10, and 100 cold-start batches**, where the distributions in each batch during cold start were **all different**. The Score distributions of each layer are **nearly identical** in all three cases. Conclusion: **The Score reflects the intrinsic characteristics of model layers and is not influenced by distribution variation**.
>
> >## **Q3 (Ablation study on $\alpha$)**
>
> We appreciate the reviewer’s constructive suggestion. We will add the results in the revised version.
>
> We conducted two sets of ablation studies on $\alpha$ with $\gamma = 0.1$, separately for clustered and non-clustered layers. In both cases, **performance is better when the SBN ratio is higher (>0.5)**. Clustered layers are sensitive to the SBN ratio **(13% change in accuracy)**, while non-clustered layers are not **(3% change)**.
>
> - Conclusion 1: **The generic knowledge (SBN) dominates during inference.**
>
> - Conclusion 2: The sensitivity to $\alpha$ is much more pronounced in layers requiring clustering than in those where clustering is unnecessary. Because TFN only provides domain-specific distributions with **limited** general knowledge distributions (as discussed in Section 2.2, Conclusion 4).
>
> - Conclusion 3: The general knowledge distribution captured by SBN is more comprehensive than that of TFN/TBN (as discussed in Section 2.2 and Appendix K).
>
> $\alpha =0.8$ for Non-Clustered Layers (TBN+SBN) and changes in [0, 0.2, 0.4, 0.6, 0.8, 1] for Clustered Layers (TFN+SBN). $\alpha$ represents the weight of SBN
> | $\alpha$ | 0    | 0.2  | 0.4  | 0.6  | 0.8  | 1    |
> |:--------:|:----:|:----:|:----:|:----:|:----:|:----:|
> | Acc/%    |57.48 |62.98 |66.81 |69.49 |70.75 |60.39 |
>
> $\alpha =0.8$ for Clustered Layers (TFN+SBN) and changes in [0, 0.2, 0.4, 0.6, 0.8, 1] for Non-Clustered Layers (TBN+SBN). $\alpha$ represents  the weight of SBN
> | $\alpha$ | 0    | 0.2  | 0.4  | 0.6  | 0.8  | 1    |
> |:--------:|:----:|:----:|:----:|:----:|:----:|:----:|
> | Acc/%    |67.20 |67.47 |68.29 |69.36 |70.75 |70.80 |
>
> >## **Q4 (The efficiency of $\alpha$-BN)**
>
> We greatly appreciate your valuable suggestion, which will make our comparison more comprehensive！
>
> The efficiency of $\alpha$-BN is consistent with that of TBN, we have provided the efficiency of TBN in Tab. 4 in paper. $\alpha$-BN only adds an interpolation step on top of TBN, whose computational overhead is negligible. We will supplement it in the revised version.
>
> >## **Q5 (Why our method  works well when batch size is small)**
>
> We appreciate your insightful question！
>
> Benefiting from our FABN module, when there is only a single sample in the batch, the sample can still provide domain-related distributions without interference from other domains. Meanwhile, SBN provides a stable and comprehensive general feature distribution. As a result, our method ensures stable inference even when bs = 1.
>
> - Why is baseline methods poor: 1. These methods are based on TBN. When the batch size is small, they do not have **stable generic knowledge (whereas we do)**. 2. Without accurate normalization, continued fine-tuning can further exacerbate model collapse.
>
> -  Why is our method good:  Setting bs = 1 is equivalent to having only one sample per cluster, where the TFN statistics degenerate to the feature statistics of that single sample (i.e., IN). At the same time, we introduce SBN to provide generic knowledge. IN of  the single simple is not affected by other domains. But the domain-related feature statistics are also less accurate due to the small sample size. Therefore, for low-resolution samples (10-C), the performance of bs = 1 is similar to bs = 128 (**a special case**). For higher-resolution samples (100-C, IN-C), the performance of bs = 1 remains stable (still outperforms other methods compared in our paper) but is inferior to bs = 128 (**the general case**).

---

> > ### Comment · Reviewer_wnbc · 2025-08-04
> >
> > Authors have addressed my concerns in the rebuttal. Therefore, I will raise my rating accordingly.

---

> > > ### Author Response · Authors · 2025-08-04
> > >
> > > we greatly appreciate the reviewer's positive response to our revision and we will polish the manuscript accordingly. Again, we sincerely thank you for raising the rating score regarding our paper.

---

### Official Review · Reviewer_kGt6 · 2025-07-01

**Clarity:** 4
**Significance:** 3
**Originality:** 3
**Rating:** 5
**Confidence:** 4

**Summary:**

This paper addresses test-time adaptation in dynamic scenarios. The authors propose FIND, a divide-and-conquer test-time normalization strategy that replaces existing batch normalization layers and is adaptable to various backbones. FIND partitions features at different layers, effectively avoiding interference between samples from different distributions. Experiments demonstrate that FIND achieves superior performance in the more complicated TTA scenarios for BN models. I believe that using the first-neighbor approach provides a sound solution to decouple features from multiple domains, which may inspire new methods for the TTA community. I thus recommend weak acceptance.

**Questions:**

1.	Why is the performance of Find* omitted from Table 2?

2.	How is accuracy measured for clustering algorithms in Table 3?

**Ethical Concerns:**

["NO or VERY MINOR ethics concerns only"]

**Final Justification:**

The authors faithfully address my concerns by providing more ablations and comparisons with SOTA. I will maintain my score of acceptance.

**Limitations:**

Please refer to the weaknesses.

**Quality:**

3

**Strengths And Weaknesses:**

**Strengths**:

1. This paper addresses a challenging scenario where test data from multiple domains are mixed in the data stream, aiming to improve the robustness of TTA in the dynamic wild world.

2. The structure of this paper is well-organized, introducing the method's design and underlying logic through observations, with a clear and well-articulated motivation.

3. The proposed method, FIND, provides a sound and nice solution to decouple features from multiple domains, thereby achieving more precise normalization for samples from different domains. Experiments also verify the superiority of Find for calibrating the BN layer at test time.

**Weaknesses**:

1. Incomplete ablation study. First, it is suggested to provide the ablation and sensitivity analysis of $\alpha$ to support the claim of mitigating the domain-specific feature perturbations. Second, it is recommended to include $\alpha$-BN and IABN in Figure 7 for more rigorous comparisons.

2. Lack of compatibility evaluation with the test-time tuning method. It is insightful to show whether the proposed method can be integrated with the test-time tuning approaches, such as Tent and ETA, to further boost the TTA performance.

3. Some important related works are missing. Since FIND aims to decouple the TTA process of different domains, it is suggested to include discussions or comparisons with the domain-aware multi-modeling TTA techniques, such as DPCore [1], BeCoTTA [2], and CoLA [3].

4. The introduction currently overlooks test-time adaptation methods built for truly “in-the-wild” settings. A brief overview of these approaches would anchor FIND within the broader TTA community, while underscoring that FIND tackles the stricter mixed-domain shift problem and is tailored for the BN models.

[1] DPCore: Dynamic Prompt Coreset for Continual Test-Time Adaptation. ICML 2025.

[2] BECoTTA: Input-dependent Online Blending of Experts for Continual Test-time Adaptation. ICML 2024.

[3] Cross-Device Collaborative Test-Time Adaptation. NeurIPS 2024.

---

> ### Author Rebuttal · Authors · 2025-07-27
>
> ## We sincerely appreciate your recognition of our motivation, novelty method and well-organized structure.  Your insightful comments are also highly valuable. Our response to your questions is elaborated below.
>
> >## **W1 (Ablation study on $\alpha$)**
>
> We thank the reviewer for  the  suggestions. This will help us further verify the sensitivity of our method to hyperparameters.
>
>  **The ablation study on $\alpha$ provides strong evidence supporting several conclusions in our paper**. We conducted two sets of ablation studies on $\alpha$ with $\gamma = 0.1$, separately for clustered and non-clustered layers. Observation: In both cases, **performance is better when the SBN ratio is higher (>0.5)**. Clustered layers are sensitive to the SBN ratio **(13% change in accuracy)**, while non-clustered layers are not **(3% change in accuracy)**.
>
> - Conclusion 1: The general knowledge (SBN) dominates during inference.
>
> - Conclusion 2: The sensitivity of $\alpha$ is much lower in layers that do not require clustering compared to those that do, because TFN only provides domain-specific distributions rather than the full general knowledge distribution (as discussed in Section 2.2, Conclusion 4).
>
> - Conclusion 3: The general knowledge distribution captured by SBN is more comprehensive than that of TFN/TBN (as discussed in Section 2.2).
>
> **$\alpha =0.8$ for Non-Clustered Layers** and changes in [0, 0.2, 0.4, 0.6, 0.8, 1] for Clustered Layers. $\alpha$ represents the weight of SBN
> | $\alpha$ | 0    | 0.2  | 0.4  | 0.6  | 0.8  | 1    |
> |:--------:|:----:|:----:|:----:|:----:|:----:|:----:|
> | Acc/%    |57.48 |62.98 |66.81 |69.49 |70.75 |60.39 |
>
> **$\alpha =0.8$ for Clustered Layers** and changes in [0, 0.2, 0.4, 0.6, 0.8, 1] for Non-Clustered Layers. $\alpha$ represents the weight of SBN
> | $\alpha$ | 0    | 0.2  | 0.4  | 0.6  | 0.8  | 1    |
> |:--------:|:----:|:----:|:----:|:----:|:----:|:----:|
> | Acc/%    |67.20 |67.47 |68.29 |69.36 |70.75 |70.80 |
>
> Good suggestions! We further evaluate the performances of  $\alpha$-BN and IABN under different batch sizes. IABN performs poorly with both small and large batch sizes. $\alpha$-BN also underperforms with large batch sizes, but shows improved performance when the batch size is 1, which is consistent with the principle of FIND.
>
> Acc (%) of CIFAR100-C, ResNet-50, BS changes in [1, 2,4, 8, 16, 64, 128]
> |         | 1     | 2     | 4     | 8     | 16    | 64    | 128   |
> |---------|-------|-------|-------|-------|-------|-------|-------|
> | IABN    | 25.58 | 26.23 | 26.38 | 26.04 | 26.00 | 26.03 | 25.80 |
> | $\alpha$-BN | 35.50 | 33.70 | 33.02 | 33.46 | 33.47 | 33.22 | 33.01 |
>
> Thanks again for your insightful suggestion. We will add the results in the revised version.
>
> >## **W2 (Combined with  test-time tuning)**
>
> We appreciate this insightful suggestions, which helps to further explore the scalability of our method.
>
>  **FIND is effective when combined with test-time tuning (with  2% improvement).**
>
> We integrated FIND with the test-time tuning methods mentioned in our paper, and obtained the following conclusions: 1. Accurate normalization is a prerequisite for successful tuning (as stated in Conclusion 1 of Section 2.2). 2. Selectively updating layers that are insensitive to domain shift leads to greater gains. The table below shows results on CIFAR10-C， CrossMix scenarios:
>
> | FIND+TNET   | $\alpha$ = 0.8 | $\alpha$ = 0.4 | $\alpha$ = 0      | $\alpha$ = 1 |
> |-------------|:-----------:|:-----------:|:--------------:|:---------:|
> | $\gamma$ = 0.1 |    72.01    |    70.15    | 67.93  |   71.03   |
> | $\gamma$ = 0.06|    72.45    |    71.16    |   69.14        |   72.37   |
>
> | FIND+EATA   | $\alpha$ = 0.8 |$\alpha$ = 0.4 | $\alpha$ = 0 | $\alpha$ = 1 |
> |-------------|:-----------:|:-----------:|:---------:|:---------:|
> | $\gamma$ = 0.1 |    72.20    |    69.98    |   67.39   |   71.14   |
> | $\gamma$ = 0.06|    72.90    |    69.96    |   68.21   |   72.7    |
>
> | FIND+DeYO   | $\alpha$ = 0.8 | $\alpha$ = 0.4 | $\alpha$ = 0 | $\alpha$ = 1 |
> |-------------|:-----------:|:-----------:|:---------:|:---------:|
> | $\gamma$ = 0.1 |    73.07    |    71.61    |   68.00   |   70.70   |
> | $\gamma$ = 0.06|    73.17    |    72.31    |   69.97   |   72.25   |
>
> Thanks again for your suggestion. We will incorporate this part in the revised version.
>
>
> >## **W3 (Compare FIND with more methods)**
>
> We are grateful for the reviewers' suggestions. Analyzing and comparing these methods can help further highlight our contributions.
>
> Our initial selection of baselines encompassed a diverse and extensive collection of SOTA methods, with a total of 10 approaches included. To address your concerns, we have identified the following works as relevant to our discussion: COME (ICLR 2025)[1], UnMix-TNS (ICLR 2024) [2], DDA (CVPR 2023)|[3], REM (ICML 2025) [4], CoLA (NeurIPS 2024)  [5].
>
> The results are shown in the table below. FIND consistently surpasses these methods by a large margin **(10%-15%)**. This further supports our conclusion that precise normalization plays a dominant role in effective TTA (Section 2.2, Conclusion 2).
>
> CoLA [5] proposes a cross-device collaborative TTA paradigm, which achieves efficient TTA by sharing and aggregating domain knowledge vectors learned during adaptation across multiple devices.  BECoTTA [6] aims to enhance the resistance to catastrophic forgetting and adaptation efficiency in the CTTA process by introducing a Mixture-of-Experts (MoE) framework. DPCore [7] aims to address continual TTA under frequent and unpredictable target domain shifts, by preventing catastrophic forgetting and negative transfer during the adaptation process.  These methods differ significantly from ours in terms of motivation. Our work specifically targets the wild world setting with dynamic data streams—where the distribution diversity within each test batch changes arbitrarily—and the strict mixed-domain scenario. To address these issues, we propose a novel "divide-and-conquer" framework (automatic distribution-aware feature disentanglement).
>
>
> Thanks again for your suggestions. we will add the descriptions and complete results of these newly given methods in the revised version. Some methods lack available code or are not compatible with our tasks or framework . We will also refer to and explain the differences in revised version.
>
> Note: The original DDA [3] paper only conducts experiments on IN-C and does not provide diffusion models for other datasets.
>
> Acc (%) of CrossMix, ResNet-50
> |           | COME [1] | UnMix-TNS [2] | DDA [3] | REM [4] | CoLA [5]  | FIND (Ours) |
> |-----------|----------|---------------|---------|---------|----------|-------------|
> | 10-C      | 60.48%   | 60.46%        | -       | 59.10%  | 62.00%     |71.54%      |
> | 100-C     | 29.76%   | 29.98%        | -       | 31.55%  | 32.52%    |40.48%      |
> | IN-C      | 16.66%   | 16.93%        | 26.10%  | 18.645% | 18.74%   |30.33%      |
>
>
> [1] COME: Test-time adaption by Conservatively Minimizing Entropy. ICLR 2025.
>
> [2] Un-Mixing Test-Time Normalization Statistics: Combatting Label Temporal Correlation. ICLR 2024.
>
> [3]  Back to the source: Diffusion-driven adaptation to test-time corruption. CVPR 2023
>
> [4] Ranked Entropy Minimization for Continual Test-Time Adaptation, ICML 2025.
>
> [5] Cross-Device Collaborative Test-Time Adaptation. NeurIPS 2024.
>
> [6] BECoTTA: Input-dependent Online Blending of Experts for Continual Test-time Adaptation. ICML 2024.
>
> [7] DPCore: Dynamic Prompt Coreset for Continual Test-Time Adaptation. ICML 2025.
>
>
> >## **W4 (Description of other work)**
>
> We greatly appreciate this valuable suggestion, as it helps to further clarify and highlight the distinctions between our approach and existing methods.
>
> **We have analyzed related wild world approaches (e.g., SAR, NOTE) in Section 4 (Related Work) and compared them in our experiments**. In the revised version, we will also mention these works in the introduction. Compared to existing approaches, our core differences are as follows:
>
> -  Motivation: We consider **fully dynamic data streams and more stringent domain mixing**. Besides the CrossMix scenario—where each batch strictly includes 15 domains—we also construct dynamic scenarios (Random and Shuffle) where each batch may contain only a single domain or a varying number of domains (2–15).
>
> - Analysis: We provide a novel analytical perspective, redefining the roles of SBN and TBN in normalization from the viewpoint of domain-related and general features.
>
> - Method: We propose a **divide-and-conquer** normalization strategy for features within a batch—not addressed by previous methods.
>
> - Result: Our method achieves high performance and robustness across all tested scenarios.
>
> >## **Q1 (The result of FIND$*$ in Table 2)**
>
> We thank the reviewer for this good question.
>
> We have provided a sensitivity analysis of $\gamma$ for Efficient-ViT in Appendix Fig. 11, and the conclusions are consistent with those for ResNet. Therefore, we did not include additional FIND* results in Table 2. We will add this content in the revised version, as shown in the table below:
> | $\gamma$ = 0.1 | 10-C   | 100-C  | IN-C   |
> |:--------------:|:------:|:------:|:------:|
> | Cross          | 78.72% | 48.30% | 29.80% |
> | Random         | 79.61% | 49.10% | 31.17% |
> | Shuffle        | 79.60% | 48.89% | 31.24% |
>
> >## **Q2 (The measure standard in Table 3)**
>
> We appreciate the reviewer for this question.
>
>  As mentioned in line 321 in our paper,  in Tab. 3, we present the **accuracy of ImageNet-C** in the CrossMix setting after replacing the LFD module in our FIND with other clustering algorithms. For a detailed comparison of our advantages over other clustering algorithms, please refer to our response to Q4 of Reviewer u3eW. In addition to the inference accuracy on the dataset, we will provide visualizations of the clustering results produced by different clustering methods at various BN layers in the revised version.

---

> > ### Comment · Reviewer_kGt6 · 2025-08-02
> >
> > The authors faithfully address my concerns by providing more ablations and comparisons with SOTA. I will maintain my score of acceptance.

---

> > > ### Author Response · Authors · 2025-08-02
> > >
> > > We greatly appreciate the reviewer’s positive response to our revision and will further polish the manuscript accordingly. Once again, we sincerely thank you for your recognition of our work.

---

### Official Review · Reviewer_u3eW · 2025-07-04

**Clarity:** 3
**Significance:** 3
**Originality:** 2
**Rating:** 4
**Confidence:** 4

**Summary:**

This paper tackles the problem of test-time adaptation in dynamic setting, where is target domain is a mixture of multiple distributions. Under this setting, standard one-size-fits-all test-time normalization techniques suffer from cross-distribution averaging. To solve this problem, this paper proposes FIND, which can captures features with similar distribution and apply different strategy to each layer. The proposed algorithm has great performance on benchmarking datasets.

**Questions:**

- What is the dataset used in Figure 3, and what are the four domains? Are they four kinds of corruptions? How generalizable this conclusion is?
- Figure 4(a): I am quite confused about the conclusion here. In line 161, the author mentioned that "SBN maintains stable performance across these variables". Although this is true, in Figure 4(a), we can see that TBN always outperforms SBN, and the variance looks smaller (especially for the dynamic setting). I don't see how this can demonstrate SBN's robustness. Figure 4(b) makes a lot of sense, although with SBN there is no adaptation of the model so it is expected to see a horizontal line of accuracy...
- I assume that "Source" means no adaptation. Why the performance is different in three difference settings? For example, in Table 2, CrossMix IN-C the accuracy is 26.05% but on Random IN-C the accuracy is 27.39%.
- Table 3 shows that the adaptation is much more effective with the proposed graph-based connecting component strategy, compared to other clustering methods. Could you provide some explanation of why this is the case? Especially given that FIND shares many similarity with some of the clustering algorithms, e.g., HDBSCAN.
- In Figure 7, why the proposed algorithm have similar performance under batch size = 1 and batch size = 128? If there is only one sample in the batch, how to do clustering and normalization? I am very confused.

**Ethical Concerns:**

["NO or VERY MINOR ethics concerns only"]

**Final Justification:**

Authors' rebuttal solves all my concerns.

**Limitations:**

Generally yes.

**Paper Formatting Concerns:**

No.

**Quality:**

3

**Strengths And Weaknesses:**

# Strength
- The solution is well-motivated. Mixture distribution is very common in real-world tasks. The author starts by observing the drop in performance, and leads to a "divide and conquer" normalization approach.
- The proposed method has great performance in corruption benchmarks
- From Figure 6, it looks like FABN outperforms TBN under a wide range of $\gamma$ (basically TBN is the proposed method with $\gamma=1$).

# Weaknesses
- The idea of interpolating between source and target statistics is very common in previous works, e.g., $\alpha$-BN. Section 2.4 basically applies $\alpha$-BN to each connected component.
- The motivation of this paper is that TBN and relevant test-time normalization algorithms assume that the target domain is homogeneous (i.e., one-size-fits-all). However, the authors ignore the fact that there are already some test-time normalization papers consider mixture of multiple distributions. One relevant paper is [1], which fits multiple sets of running statistics for each normalization layer. However, the author didn’t mention this, or compare the proposed method to this paper.

[1] Un-Mixing Test-Time Normalization Statistics: Combatting Label Temporal Correlation. ICLR 2024.

# Minor
- Equation 1: L is used for both spatial dimension (H*W) and its iterator. Though this does not affect understanding, typically one should use different notation.
- $\gamma$ is used both in Equation (7) and as the threshold of score in line 242. This might be a little bit confusing.
- Line 271: detalis -> details

---

> ### Author Rebuttal · Authors · 2025-07-27
>
> ## Thank you for recognizing our well-motivated solution and strong empirical results. Your great comments provide constructive guidance for the further exploration of our work. Our response to your concerns is presented as follows.
>
>
>
> >## **W1 (Difference in interpolating)**
>
> We thank the reviewer for this insightful question, which allows us to further highlight the contributions of our method.
>
> We would like to clarify that **the target, purpose, and core idea of our interpolating  are fundamentally different from previous methods, despite any superficial similarities in form**.
>
> - Core idea: the core of our method lies in the entire "divide-and-conquer" framework, where the "divide" step is critical. Interpolating  serves as an efficient means for "conquering" after precise partitioning. **Without an accurate partitioning of domain-related features, interpolating  is meaningless**. Therefore, traditional interpolation methods such as $\alpha$-BN are ineffective in dynamic environments.
>
> - Interpolating  target: the interpolating  target of our FABN is no longer the traditional TBN (the entire batch) or IN (a single sample), but rather the TFN (multiple clusters) obtained through distribution-aware partitioning. TFN can adaptively capture the distribution diversity within a batch, which TBN and IN are unable to achieve.
>
> - Purpose of interpolating: We leverage SBN to complement the general features missing from each TFN, thus constructing a complete feature distribution (Section 2.2, Conclusion 4). The interpolation in $\alpha$-BN or IABN is intended to reduce distribution gaps.
>
> - New theoretical insights and perspectives: We provide a novel analysis of the roles of SBN and TFN, with detailed studies of general/domain-related features in Section 2.2 and Appendix K.
>
> - Results: Our divide-and-conquer based interpolating  consistently outperforms $\alpha$-BN and IABN in both performance and robustness across various scenarios.
>
> >## **W2 (Method comparison)**
>
> We thank the reviewer for this good suggestion.
>
> **NOTE ([6] in our paper) and UnMix-TNS [1] are highly similar in both motivation and methodology, and we have analyzed and compared NOTE with our method in the paper.  Notably, there are substantial differences between UnMix-TNS and our approach.**
>
> - Motivation: UnMix-TNS is designed to address **non-i.i.d** (label imbalance) scenarios (although they tested mixed domain scenarios in their experiments, this is not their main focus), while our paper specifically targets **dynamic data streams and strict mix domain settings in wild world.** In addition to the strict mix-domain scenario (where each batch contains a mixture of 15 domains), we also consider fully random and dynamic settings—where each batch may contain only a single domain or any number of domains (2–15).
>
> - Method:  Although UnMix-TNS replicates K SBNs using Gaussian perturbations, these SBNs operate as an **integrated whole** and thus cannot effectively capture multiple domain distributions in mixed-domain scenarios. In contrast, our method explicitly decomposes the computation of BN statistics into multiple clusters along the feature dimension in a layer-wise manner, thereby achieving a genuine **"divide-and-conquer"** approach that previous methods lack.
>
> As shown in the following table, our approach outperforms UnMix-TNS by a significant margin **(11%-15%)** in mixed domain settings. We will include these results and cite UnMix-TNS in the revised version. We hope this clarification addresses your concerns. Thank you for your valuable feedback.
>
> | Dataset   | FIND (ours) | UnMix-TNS |
> |-----------|-------------|-----------|
> | 10-C      | 71.54%      | 60.46%    |
> | 100-C     | 40.48%      | 29.98%    |
> | IN-C      | 30.33%      | 16.93%    |
>
> [1] Un-Mixing Test-Time Normalization Statistics: Combatting Label Temporal Correlation. ICLR 2024.
>
> >## **Q1 (Details of Figure 3 in paper)**
>
>
> The dataset used is **CIFAR100-C**, with domains corresponding to four types of corruptions: **Gaussian Noise, Zoom Blur, Snow, and Pixelate**, which represent the **real-world categories of noise, blur, weather, and digital**, respectively. **This conclusion is general and reflects the intrinsic characteristics of the models**; it also holds on the other two datasets and with the ViT architecture. We will provide additional visualizations for different datasets and architectures in the revised version.
>
>
> >## **Q2 (Explain of Figure 4(a))**
>
> Thank you for your questions, this will help us further improve the descriptions in the manuscript.
>
> In Fig. 4(a),  **“stable” refers to SBN is unaffected by the number of domains in a batch, while TBN is influenced by it.**
>
> - Phenomenon:  as the number of domains in the test data increases (from 1 to 15), the performance of SBNs **remains unchanged** in both static and dynamic settings, whereas the performance gap between the two TBNs **continues to widen**. This indicates that SBN is not affected by domain diversity within a batch, whereas TBN is highly sensitive to it. Therefore, SBN provides stable and reliable general features.
>
> - Additional explanation: TBN consistently outperforms SBN because SBN does not contain any domain-specific features. This does not contradict the fact that SBN is unaffected by the number of domains in a batch.
>
>
> >## **Q3 (Different source results)**
>
> Excellent question!
>
>  **The data streams for the Random and Shuffle settings are longer than for CrossMix**. In CrossMix, each batch contains a mixture of 15 domains, while  the other two settings include batches from a single domain as well as mixtures of varying domain numbers (from 2 to 15). To ensure all these cases are sufficiently represented, we **extend the data stream length for Random and Shuffle to 1.5 times that of CrossMix**. Specifically, after one complete pass through the dataset, we randomly sample an additional 50% of the data for further input. Although the data stream lengths differ across settings, **the comparisons between methods are entirely fair**. We will clarify and supplement this detail in the revised version.
>
>
> >## **Q4 (Clustering difference)**
>
> Thank you for this excellent question!
>
> FIND achieves **fully automatic perception and dynamic grouping without requiring any additional parameters**. Moreover, FIND possesses **the capability for fine-grained local perception**. But other clustering methods do not possess these capabilities.
>
> - Adaptation to dynamic distributions: FIND adopts a "nearest neighbor connectivity" principle forming connected components. This enables FIND to adaptively capture the true structure of the data distribution, making it highly robust to complex shapes and unknown numbers of clusters. In contrast, methods like K-means and HDBSCAN rely on preset cluster numbers, global density, or hyperparameters,  making them **difficult to generalize to highly dynamic distributions**, and requiring frequent parameter tuning.
>
> - Adaptation to complex distributions: By focusing on local neighbor search, FIND achieves **fine-grained clustering** and is sensitive to subtle local structures in both high-and low-density regions. Therefore, FIND can handle elongated, ring-shaped, or nested distribution structures and automatically distinguish distributions with unknown numbers or highly imbalanced sizes. Even under overlap, FIND can identify reasonable clusters based on local similarity. By comparison, K-means and HDBSCAN are mainly suitable for **spherical, balanced distributions**—while HDBSCAN can handle some complex cases, it is highly sensitive to parameters and unsuitable for real-time or large-scale scenarios. Moreover, when high-density and low-density clusters coexist, HDBSCAN is prone to misclassifying low-density clusters.
>
> We illustrate these points with two examples:
>
> - Example 1: When domains are completely separated without any overlap, traditional clustering algorithms (e.g., HDBSCAN) and FIND can all separate them successfully.
>
> - Example 2: When domains are mostly separated but overlap at the boundaries or one domain encloses another (as shown in Fig. 3(a) in our paper), FIND can provide finer-grained perception and segmentation for the overlapping or enclosing parts. For instance, red domain samples surrounded by green domain examples can still be assigned to the correct cluster via nearest neighbor search. However, HDBSCAN and K-means only consider global density or distance, leading to hard splits at the overlap and resulting in unstable normalization.
>
>
> Conclusion:  FIND achieves **fully automatic perception and dynamic grouping**. Other clustering algorithms **depend on hyperparameters or  can only partition the space globally, lacking the capability for fine-grained perception in complex scenarios**.
>
>
> >## **Q5 (Similar performances of small and large batch size)**
>
> Thank you for this great observation!
>
>
> **The similar performance between bs = 1 and bs = 128 in Fig. 7(a) is a special case. Under general circumstances, increasing the batch size  improves the performance of FIND (as shown in Fig. 15 in the Appendix)**.
>
> - Reasons for special case: **The  stable capture of domain-specific features (Conclusions 2 and 4 in Section 2.2)**.  Setting bs = 1 is equivalent to having only one sample per cluster, where the TFN statistics degenerate to the feature statistics of that single sample (i.e., IN).  The domain-related feature of the single sample is not affected by other domains.  Therefore, for **low-resolution samples** (10-C), the performance of bs = 1 is similar to bs = 128.
>
> - Reasons for general case: The domain-related feature statistics are less accurate due to the small sample size (bs = 1) and will be disturbed by the style of the sample itself. Therefore, for **higher-resolution** samples (100-C, IN-C), performance remains stable when bs = 1  (still outperforms other methods compared in our paper) but is inferior to bs = 128.

---

> ### Author Response · Authors · 2025-08-08
>
> Dear reviewer, thank you for your positive evaluation of our work. We sincerely appreciate your thoughtful suggestions and will add these additional results to the revision. Please let us know if you need more information. We greatly appreciate this opportunity to improve our work.

---

> > ### Comment · Reviewer_u3eW · 2025-08-08
> >
> > Thanks for your rebuttal!
> >
> > The explaination is detailed and straightforwad. I definitely had a misunderstanding regarding Q2: the robustness is w.r.t. static vs. dynamic, not w.r.t. batch size.
> >
> > I think it is a good paper and should be accepted. Good luck!

---

> > > ### Author Response · Authors · 2025-08-09
> > >
> > > We greatly appreciate the reviewer’s positive response to our revision and will further polish the manuscript accordingly. Once again, we sincerely thank you for your recognition of our work.

---

> ### Comment · Area_Chair_rHBB · 2025-08-08
>
> Hi Reviewer u3eW,
>
> The author–reviewer discussion phase will close soon. If you have not yet participated, please take this opportunity to review the rebuttal, check how your comments have been addressed, and share any remaining concerns with the authors. Your engagement in these final days is key to ensuring a thorough review process.
>
> AC

---

### Comment · Area_Chair_rHBB · 2025-08-04
**Reminder: Review Rebuttal and Submit Final Justification**

Dear Reviewers,

As we approach the end of the author–reviewer discussion phase (**Aug 6, 11:59pm AoE**), I kindly remind you to read the author rebuttal carefully, especially any parts that address your specific comments. Please consider whether the response resolves your concerns, and if not, feel free to engage in further discussion with the authors while the window is still open.

Your timely participation is important to ensure a fair and constructive review process. If you feel your concerns have been sufficiently addressed, you may also submit your Final Justification and update your rating early. Thank you for your contributions.

Best,

ACs

---

> ### Author Response · Authors · 2025-08-09
>
> We sincerely thank all the reviewers for their valuable time and effort in reviewing our paper. We also appreciate the chairs’ responsible management and effective communication throughout the review process.
>
> As a summary of the discussion phase,
>
> ## **During our discussions with the reviewers:**
>
> - We validated the effectiveness of combining  our FIND with fine-tuning to address  the concerns raised by reviewer kGt6.
>
> - We elaborated on the key differences between our work and previous studies to address similar concerns from reviewers u3eW and dD3f. Furthermore, in addition to the original 10 baselines, we compared our method with more related works. The experimental results demonstrate that FIND significantly outperforms existing methods, which addresses the concerns raised by reviewers kGt6 and 4ioQ.
>
> - We further validated the high efficiency of our  FIND during inference to address the concerns raised by reviewer 4ioQ.
>
> - We specifically elaborated on the core ideas and functionalities of our method, and demonstrated the effectiveness of each component to address the concerns raised by reviewer wnbc.
>
> - We provided targeted explanations and engaged in detailed discussions regarding all remaining concerns.
>
> ## **We are encouraged by all the reviewers’ positive feedback:**
>
> We deeply appreciate the positive feedback from the reviewers that our work has good motivation, an innovative approach, strong empirical results, and a well-organized structure. We are pleased that, after addressing all the questions and concerns, all reviewers confirmed that their concerns had been fully resolved.
>
> As a result, we successfully gained the approval of all five reviewers. We sincerely thank the reviewers for their insightful suggestions and invaluable contributions to improving the quality of our submission.

---

### Decision · Program_Chairs · 2025-09-17

**Decision:**

Accept (poster)

**Comment:**

All reviewers have reached a consensus to accept this paper. After carefully reading the manuscript, the reviews, and the author rebuttal, I agree with this assessment. The paper addresses test-time adaptation under dynamic, mixed-distribution scenarios by proposing a novel divide-and-conquer normalization framework (FIND), which includes Layer-wise Feature Disentanglement (LFD), Feature-Aware Batch Normalization (FABN), and Selective FABN (S-FABN). The method is well-motivated, technically sound, and empirically validated across diverse architectures and datasets. The authors provided thorough responses during the rebuttal, clarifying key differences from related work, adding new ablations and comparisons, and demonstrating generalizability. I recommend acceptance and encourage the authors to further polish the paper and incorporate the reviewers' suggestions in the camera-ready version.